# Diffusion Twigs with Loop Guidance for Conditional Graph Generation

**Giangiacomo Mercatali** [†] [*]
HES-SO Genève
University of Manchester
giangiacomo.mercatali@hesge.ch

**Yogesh Verma** [†]
Aalto University
yogesh.verma@aalto.fi

**Andre Freitas**
Idiap Research Institute
University of Manchester
NBC, CRUK Manchester Institute
andre.freitas@idiap.ch

**Vikas Garg**
YaiYai Ltd & Aalto University
vgarg@csail.mit.edu

## Abstract

We introduce a novel score-based diffusion framework named *Twigs* that incorporates multiple co-evolving flows for enriching conditional generation tasks. Specifically, a central or *trunk* diffusion process is associated with a primary variable (e.g., graph structure), and additional offshoot or *stem* processes are dedicated to dependent variables (e.g., graph properties or labels). A new strategy, which we call *loop guidance*, effectively orchestrates the flow of information between the trunk and the stem processes during sampling. This approach allows us to uncover intricate interactions and dependencies, and unlock new generative capabilities. We provide extensive experiments to demonstrate strong performance gains of the proposed method over contemporary baselines in the context of conditional graph generation, underscoring the potential of Twigs in challenging generative tasks such as inverse molecular design and molecular optimization. Code is available at https://github.com/Aalto-QuML/Diffusion_twigs.

## 1 Introduction

Conditional graph generation is a fundamental problem in scientific domains such as *de novo* drug design [21, 43, 74] and material design [39]. However, searching for new molecules with desired physicochemical properties poses significant challenges to traditional brute-force methods due to the vast combinatorial spaces [64]. With the advent of neural networks [44], deep generative models have emerged as a powerful tool for learning informative conditional representations of molecules, facilitating the development of *in silico* methods for chemical design [16, 31, 61, 73].

Score-based diffusion generative models (SGMs) and denoising probabilistic diffusion models (DDPMs) [24, 67] have recently emerged as powerful techniques for training deep networks on graph-structured data, with applications spanning molecular design [37, 53, 36, 81], molecular docking [6], molecular dynamics simulations [78], protein folding [79], and backbone modeling [70]. Notably, diffusion models exhibit superior capabilities for *conditional* graph generation, excelling in both discrete [26, 75, 49] and continuous [3, 28, 45, 11] settings. The training of the mentioned conditional diffusion models is achieved by two types of diffusion guidance algorithms: *classifier-based guidance*

---

[†]Equal Contribution. Order decided via coin flip.
[*]Work done while at the University of Manchester

38th Conference on Neural Information Processing Systems (NeurIPS 2024).

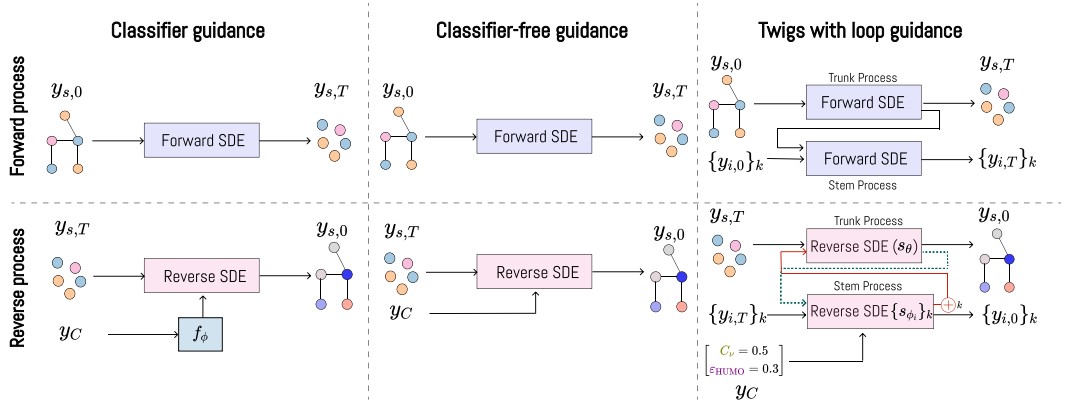

Figure 1: **Overview of the proposed method (Twigs).** We define two types of diffusion processes: (1) multiple *Stem* processes ($s_{\phi_i}$), which unravel the interactions between graph structure and single properties, and (2) the *Trunk* process, which orchestrates the combination of the graph structure score from $s_\theta$ with the stem process contributions from $s_{\phi_i}$. During the forward process, the structure $\mathbf{y}_s$ and the properties $\{\mathbf{y}_i\}_k$ co-evolve toward noise. In each step of the reverse process, the structure is first denoised and subsequently used to denoise the properties (indicated by the green-dashed line). Such de-noised properties are then utilized, in turn, to further denoise the structure (red line), in a process that resembles a *guidance loop*.

[8], which involves training a separate property predictor model alongside the diffusion model; and *classifier-free guidance* [23], which integrates scores from both unconditional and conditional diffusion models. While these guidance techniques have been found to be effective, the algorithm design is not tailored to encompass the intricate hierarchical or multi-resolution elements inherent in conditional generation. Consequently, it is plausible that this inadequacy may contribute to suboptimal representations, particularly notable in tasks such as conditional graph generation. The recent success of hierarchical diffusion flows in various domains, such as modeling interactions between node and edge features [37], multi-resolution modeling [25], decision-making [47], and conditional image generation [4, 71] underscores the need to integrate hierarchical information beyond the capabilities of classifier-based and classifier-free guidance.

We assert that conditional diffusion models for structured spaces, such as graphs, could be enhanced with *hierarchical conditional processes*. Specifically, rather than treating heterogeneous structural and label information uniformly within the hierarchy, we advocate for the co-evolution of multiple processes with *distinct roles* (asymmetric). These roles encompass a primary process governing the structural evolution alongside multiple secondary processes responsible for driving conditional content. We aim to propose an alternative to existing conditional graph diffusion techniques (outlined in Table 1) by bestowing the models with finer control over two key aspects: 1) the evolution of structural graph components, including nodes and edges, and 2) the co-adaptation of the graph structure in conjunction with one or more associated properties.

Towards this objective, we present a novel diffusion framework for conditional generation named Twigs, drawing analogies from the trunk and offshoots of a tree. Concretely, we establish a central *trunk* process governing a primary variable, which interacts with several *stem* processes, each associated with a secondary variable. In contrast with classifier-free and classifier-based methodologies, a novel conditional mechanism, termed *loop guidance*, orchestrates information exchange between the trunk and the stem processes (refer to Figure 1). Our methodology facilitates the acquisition of flexible representations, capitalizing on the disentanglement of intricate interactions and dependencies. We formalize our framework by drawing upon the theory of denoising score matching [67] and leveraging tools derived from stochastic differential equations (SDEs) [1]. The effectiveness of Twigs is substantiated through compelling empirical validation across various conventional constrained generation tasks, utilizing both molecular and generic graph datasets.

## 1.1 Contributions

In summary, this paper makes the following key contributions:

Table 1: **Comparison of related methodologies**. `Twigs` is the first method that enables a seamless orchestration of multiple asymmetric property-oriented hierarchical diffusion processes via SDEs.

| Method | Conditional | Asymmetric | Multiple flows | Continuous (SDEs) |
|---|---|---|---|---|
| GDSS [37] | ✗ | ✗ | ✓ | ✓ |
| EEGSDE [3] | ✓ | ✗ | ✗ | ✓ |
| MOOD [45] | ✓ | ✗ | ✗ | ✓ |
| JODO [28] | ✓ | ✗ | ✗ | ✓ |
| EDGE [5] | ✗ | ✗ | ✓ | ✗ |
| GraphMaker [46] | ✓ | ✗ | ✓ | ✗ |
| Nisonoff et al. [52] | ✓ | ✗ | ✗ | ✗ |
| Gruver et al. [18] | ✓ | ✗ | ✗ | ✓ |
| Klarner et al. [40] | ✓ | ✓ | ✗ | ✗ |
| Twigs (ours) | ✓ | ✓ | ✓ | ✓ |

- **(Conceptual and methodological)** The introduction of a new score-based, end-to-end trainable, non-autoregressive generative model `Twigs` designed for acquiring conditional representations. Our approach enables precise guidance of multiple property-conditioned diffusion processes.

- **(Technical)** We present a robust mathematical framework, including a novel strategy called *loop guidance*, that employs tools from Stochastic Differential Equations (SDEs) to derive both the forward diffusion process and its corresponding reverse SDE for conditional generation. This framework is designed to seamlessly integrate additional contexts as conditioning information.

- **(Empirical)** We showcase the versatility of the proposed diffusion mechanism (`Twigs`) through extensive empirical evidence across various challenging conditional graph generation tasks, consistently surpassing contemporary baselines.

## 2 Related works

In Table 1 we provide an overview of the similarities and differences between `Twigs` and related methods. We refer the reader to Appendix E for additional related work.

**Diffusion guidance** is typically applied to regulate the diffusion process for conditional generation. Previous approaches that perform class-conditional generation are divided into classifier-based [8], and classifier-free guidance [23]. While some works model diffusion with multiple flows [5, 37, 46], they treat nodes and edges in a symmetric way; i.e., they associate multiple flows for nodes and edges that have equivalent contributions (in other words, these flows have the same roles). We instead abstract graph properties as secondary processes that branch from, and interact with, the main process that pertains to the graph structure. In addition, while other guidance methods are related [18, 40, 52], they do not leverage multiple diffusion flows. To our knowledge, the proposed method is the first to incorporate multiple diffusion flows in a hierarchical fashion for conditional generation. We formalize in Table 2 how `Twigs` differs, mathematically, from classifier-free and classifier-based methods.

**Conditional Diffusion for Graphs** Recent advancements in generative modeling have prominently featured score-based techniques (SGM), utilizing diffusion or stochastic differential equations (SDEs) [19, 32, 35, 37, 48], including for graph generation [3, 5, 13, 14, 15, 18, 26, 40, 45, 46, 52, 72, 75, 82]. Guidance methods have been adopted in conditional molecule generation settings. The works from Hoogeboom et al. [26], Huang et al. [28, 28], Xu et al. [82] are classifier-free approaches, while Bao et al. [3], Vignac et al. [75], Lee et al. [45] focus on classifier-based methods. Diverging from these approaches, we explicitly model the dynamic interaction between primary variables (e.g., graph structure) and dependent variables (e.g., graph properties) using dedicated diffusion processes to achieve more expressive representations and improve performance for conditional generation.

## 3 Diffusion Twigs

**Method overview** We extend score-based techniques [67] for training conditional diffusion models over graphs. Differently from current guidance methods, as summarised in Table 2, we leverage a finer control over the structure and graph properties to diffuse multiple hierarchical processes, toward achieving a more robust representation. Our method, `Twigs`, defines a *trunk* process over the primary

Table 2: `Twigs` comparison to Classifier-based [8] and Classifier-free [23] guidance, applied for conditional generation in Diffusion models. Here $\mathbf{y}_s$ represents the graph structure, $\{\mathbf{y}_i\}_k$ represent the $k$-properties of graph. The $f_\phi$ function is the classifier, $\epsilon_\theta$ and $s_{\theta,\phi}$ are learnable score models.

| Method | Diffusion Scheme | Approach |
|---|---|---|
| Class. based | $d\mathbf{y}_s = \mathbf{f}(\mathbf{y}_{s,t},t)dt + g(t)d\mathbf{w}$ 
 $d\mathbf{y}_s = [\mathbf{f}(\mathbf{y}_{s,t},t) - g_t^2 \nabla_{\mathbf{y}_{s,t}} \log p_t(\mathbf{y}_{s,t}, \{\mathbf{y}_i\}_k)]dt + g_t d\bar{\mathbf{w}}$ | $\nabla_{\mathbf{y}_{s,t}} \log p(\mathbf{y}_{s,t}, \{\mathbf{y}_i\}_k) = \nabla_{\mathbf{y}_{s,t}} \log p(\mathbf{y}_{s,t}) + \nabla_{\mathbf{y}_{s,t}} \log p(\{\mathbf{y}_i\}_k \mid \mathbf{y}_{s,t})$ 
 $\approx -\frac{1}{\sqrt{1-\bar{\alpha}_t}} \epsilon_\theta(\mathbf{y}_{s,t}) + \nabla_{\mathbf{y}_{s,t}} \log f_\phi(\{\mathbf{y}_i\}_k \mid \mathbf{y}_{s,t})$ |
| Class. free | $d\mathbf{y}_s = \mathbf{f}(\mathbf{y}_{s,t},t)dt + g(t)d\mathbf{w}$ 
 $d\mathbf{y}_s = [\mathbf{f}(\mathbf{y}_{s,t},t) - g_t^2 \nabla_{\mathbf{y}_{s,t}} \log p_t(\mathbf{y}_{s,t}, \{\mathbf{y}_i\}_k)]dt + g_t d\bar{\mathbf{w}}$ | $\nabla_{\mathbf{y}_{s,t}} \log p(\{\mathbf{y}_i\}_k \mid \mathbf{y}_{s,t}) = \nabla_{\mathbf{y}_{s,t}} \log p(\mathbf{y}_{s,t} \mid \{\mathbf{y}_i\}_k) - \nabla_{\mathbf{y}_{s,t}} \log p(\mathbf{y}_{s,t})$ 
 $= -\frac{1}{\sqrt{1-\bar{\alpha}_t}} (\epsilon_\theta(\mathbf{y}_{s,t}, \{\mathbf{y}_i\}_k) - \epsilon_\theta(\mathbf{y}_{s,t},t))$ |
| Twigs | $d\mathbf{y}_s = \mathbf{f}(\mathbf{y}_{s,t},t)dt + g(t)d\mathbf{w}, \{d\mathbf{y}_i\}_k = \mathbf{f}(\mathbf{y}_{s,t}, \mathbf{y}_{i,t},t)dt + g(t)d\mathbf{w}$ 
 $d\mathbf{y}_s = [\mathbf{f}(\mathbf{y}_{s,t},t) - g_t^2 \nabla_{\mathbf{y}_{s,t}} \log p_t(\mathbf{y}_{s,t}, \{\mathbf{y}_{i,t}\}_k)]dt + g_t d\bar{\mathbf{w}}$ 
 $\{d\mathbf{y}_i\}_k = [\mathbf{f}(\mathbf{y}_{s,t}, \mathbf{y}_{i,t},t) - g_t^2 \nabla_{\mathbf{y}_{i,t}} \log p_t(\mathbf{y}_{s,t}, \mathbf{y}_{i,t})]dt + g_t d\bar{\mathbf{w}}$ | $\nabla_{\mathbf{y}_{s,t}} \log p_t(\mathbf{y}_{s,t}, \{\mathbf{y}_{i,t}\}_k) = \nabla_{\mathbf{y}_{s,t}} \log p_t(\mathbf{y}_{s,t}) + \sum_i \nabla_{\mathbf{y}_{s,t}} \log p_t(\mathbf{y}_{i,t} \mid \mathbf{y}_{s,t})$ 
 $\nabla_{\mathbf{y}_{s,t}} \log p_t(\mathbf{y}_{s,t}) \approx s_{\theta,t}(\mathbf{y}_{s,t}), \nabla_{\mathbf{y}_{s,t}} \log p_t(\mathbf{y}_{i,t} \mid \mathbf{y}_{s,t}) \approx s_{\phi,t}(\mathbf{y}_{s,t}, \mathbf{y}_{i,t})$ 
 $\nabla_{\mathbf{y}_{s,t}} \log p_t(\mathbf{y}_{s,t}, \{\mathbf{y}_{i,t}\}_k) = s_{\theta,t}(\mathbf{y}_{s,t}) + \sum_i s_{\phi,t}(\mathbf{y}_{s,t}, \mathbf{y}_{i,t})$ |

variable (graph structure) $\mathbf{y}_s$, and a *stem* process over each dependent variable $\mathbf{y}_i \in \mathbb{R}$ (e.g., graph property). We achieve the desired flexibility with a variable $\mathbf{y}_s$ that encompasses both node features and the adjacency matrix as well as the coordinates. The details of the dimensions of $\mathbf{y}_s$ are given in Section B.1 for the 3D case, and in Section B.2 for the 2D case.

**Forward process**   We define multiple forward processes within a hierarchy that co-evolves data and properties into noise. The *trunk* forward process for the graph structure $\mathbf{y}_s$ is defined as

$$d\mathbf{y}_s = \mathbf{f}_s(\mathbf{y}_{s,t}, t)dt + g_s(t)d\mathbf{w} \tag{1}$$

where $\mathbf{f}_s$ and $g_s$ are corresponding diffusion and drift functions, and $d\mathbf{w}$ is the Wiener noise. The *stem* forward process over the $k$ dependent variables $\mathbf{y} = \{\mathbf{y}_1, \ldots, \mathbf{y}_k\}$ is defined as

$$d\mathbf{y}(t) = \begin{pmatrix} d\mathbf{y}_1(t) \\ \vdots \\ d\mathbf{y}_k(t) \end{pmatrix} = \begin{pmatrix} \mathbf{f}_p(\mathbf{y}_{1,t}, \mathbf{y}_{s,t}, t)dt + g_p(t)d\mathbf{w} \\ \vdots \\ \mathbf{f}_p(\mathbf{y}_{k,t}, \mathbf{y}_{s,t}, t)dt + g_p(t)d\mathbf{w} \end{pmatrix} \tag{2}$$

Here, $\mathbf{f}_p$ and $g_p$ denote the diffusion and drift functions, respectively, for the $k$ stem processes. Collectively, along with the trunk forward process, they constitute `Twigs`. These operations introduce random Gaussian noise, iteratively, to the data toward a prior (typically Gaussian) distribution.

**Reverse Process**   The `Twigs` reverse process starts from the prior distribution (Gaussian noise) towards the data distribution. A key difference with Song et al. [67] is that here our variable $\mathbf{y}_t$ comprises both structure *and properties*, leading to the following modification of the overall diffusion process:

$$d\mathbf{y}_t = [f(\mathbf{y}_t, t) - g_t^2 \nabla_{\mathbf{y}_t} \log p_t(\mathbf{y}_t)]dt + g_t d\bar{\mathbf{w}} \qquad \text{where} \quad \mathbf{y}_t = \{\mathbf{y}_{s,t}, \{\mathbf{y}_{i,t}\}_{i=1}^k\} . \tag{3}$$

We derive Equation (3) in Section A.1. The joint distribution over the trunk and stem processes is assumed to factorize as

$$p_t(\mathbf{y}_{s,t}, \mathbf{y}_{1,t}, ..., \mathbf{y}_{k,t}) = p_t(\mathbf{y}_{s,t}) \prod_{i=1}^k p_t(\mathbf{y}_{i,t} \mid \mathbf{y}_{s,t}) . \tag{4}$$

In turn, the score function simplifies as in Equation (5), leading to the decomposition in Equation (6).

$$\nabla_{\mathbf{y}_t} \log p_t(\mathbf{y}_{s,t}, \mathbf{y}_{1,t}, \ldots, \mathbf{y}_{k,t}) = \nabla_{\mathbf{y}_t} \log p_t(\mathbf{y}_{s,t}) + \sum_{i=1}^k \nabla_{\mathbf{y}_t} \log p_t(\mathbf{y}_{i,t} \mid \mathbf{y}_{s,t}) \tag{5}$$

$$d\mathbf{y}_t = [\mathbf{f}(\mathbf{y}_t, t) - g_t^2 (\nabla_{\mathbf{y}_t} \log p_t(\mathbf{y}_{s,t}) + \sum_{i=1}^k \nabla_{\mathbf{y}_t} \log p_t(\mathbf{y}_{i,t} \mid \mathbf{y}_{s,t}))]dt + g_t d\bar{\mathbf{w}} \tag{6}$$

**Conditional modeling**   We expand our proposed approach to enable conditional generation with an external context $\mathbf{y}_C = \{\mathbf{y}_c \mid c \in C\}$, where $C \subseteq \{1, \ldots, k\}$. The context can be represented as a scalar or vector, describing a particular value associated with a data-dependent variable. For example, in case of molecules, it could represent one or more of the $k$ properties such as the Synthetic Accessibility (SA) score or the Quantitative Estimate of Drug likeness (QED). This extension modifies the joint distribution for the score function in Equation (5).

> **Reverse SDE under conditioning context**
>
> The reverse SDE for $\mathbf{y}_t = \{\mathbf{y}_{s,t}, \{\mathbf{y}_{i,t}\}_k\}$ give an external conditioning context $\mathbf{y}_C$ is shown below (details in Appendix A.2).
>
> $$\mathrm{d}\mathbf{y}_t = [\mathbf{f}(\mathbf{y}_t, t) - g_t^2 \nabla_{\mathbf{y}_t} \log p_t(\mathbf{y}_t, \mathbf{y}_C)]\mathrm{d}t + g_t \mathrm{d}\bar{\mathbf{w}} \tag{7}$$

We resort to the following factorization of the distribution, conditioned on the context $\mathbf{y}_C$:

$$p_t(\mathbf{y}_{s,t}, \{\mathbf{y}_{i,t}\}_k, \mathbf{y}_C) = \prod_i^k p_t(\mathbf{y}_{i,t} \mid \mathbf{y}_{s,t}, \mathbf{y}_C) p_t(\mathbf{y}_{s,t}, \mathbf{y}_C)$$

As a result, the factorization of the score function $\nabla_{\mathbf{y}_t} \log p_t(\mathbf{y}_{s,t}, \{\mathbf{y}_{i,t}\}_k, \mathbf{y}_C)$ amounts to

$$\nabla_{\mathbf{y}_t} \log p_t(\mathbf{y}_{s,t}, \mathbf{y}_C) + \sum_{i \notin C}^k \nabla_{\mathbf{y}_t} \log p_t(\mathbf{y}_{i,t} \mid \mathbf{y}_{s,t}) + \sum_c^C \sum_i^k \delta_{i=c} \nabla_{\mathbf{y}_t} \log p_t(\mathbf{y}_{i,t} \mid \mathbf{y}_{s,t}, \mathbf{y}_c) \tag{8}$$

The above-factorized score function parameterizes our reverse diffusion process, thus offering a novel approach to integrate external contextual information into conditional generation.

**Training** We propose to train `Twigs` by incorporating the factorization from Equation (8) within a score-matching objective function [30, 67]. Algorithm 1 shows the training procedure to learn two types of time-dependent score-based models: $s_{\theta,t}$, which approximates the trunk variable, and $s_{\phi_i,t}$ which approximates the coupling between the stem variable and the trunk variable. The objective function for optimizing the score networks $s_\theta, s_{\phi_i}$, is given as follows:

$$\min_{\theta,\phi_i} \mathbb{E}_t \left\{ \lambda_{\mathbf{y}_t}(t) \mathbb{E}_{\mathbf{y}_0} \mathbb{E}_{\mathbf{y}_t \mid \mathbf{y}_0} \| s_{\theta,t}(\mathbf{y}_{s,t}, \mathbf{y}_c) + \sum_i^k s_{\phi_i,t}(\mathbf{y}_{i,t}, \mathbf{y}_{s,t}, \mathbf{y}_c) - \nabla_{\mathbf{y}_t} \log p_t(\mathbf{y}_t, \mathbf{y}_C) \|_2^2 \right\} \tag{9}$$

where $\mathbb{E}_{\mathbf{y}_0} = \mathbb{E}_{\mathbf{y}_{s,0}, \mathbf{y}_{i,0}}$ and $\mathbb{E}_{\mathbf{y}_t} = \mathbb{E}_{\mathbf{y}_{s,t}, \mathbf{y}_{i,t}}$. It is worth noting that the influence introduced by the variable $s_{\phi_i}$ provides the directions for the diffusion model to converge into distributions with the desired properties. Such property-oriented knowledge operates in conjunction with the structural information provided by $s_\theta$, resulting in a novel form of guidance that is orchestrated by a branching diffusion process, named *Loop guidance*.

| **Algorithm 1** Training `Twigs` | **Algorithm 2** Generating with `Twigs` |
|---|---|
| **Input:** Dataset $\mathcal{D}$, iterations $n_{\text{iter}}$, batch size $B$, number of batches $n_B$, $K$ properties to consider Initialize parameters $s_{\theta,t}, \{s_{\phi_i,t}\}_{i=1}^K$ for Score Networks | **Input:** Score-based models $s_{\theta,t}, \{s_{\phi_i,t}\}_{i=1}^K$, Time step schedule $\{t\}_{t=T}^0$, Langevin MCMC step size $\alpha$, External context $\boldsymbol{y}_C$ |
| **for** $k = 1, \ldots, n_{\text{iter}}$ **do** | |
|   **for** $b = 1, \ldots, n_B$ **do** | |
|     $t \sim \mathcal{U}(0, 1]$ | $\boldsymbol{y}_{s_T}, \{\boldsymbol{y}_{i,T}\}_{i=1}^K \sim \mathcal{N}(0, I)$ |
|     $\mathcal{D}_b = \{(\boldsymbol{y}_{s,l}, \{\boldsymbol{y}_{i,l}\}_{i=1}^K)_{l=1}^B, \boldsymbol{y}_C\} \sim \mathcal{D}$ | **for** $t = T, \ldots, 0$ **do** |
|     $\mathcal{L}_b \leftarrow$ Eq. 9 |   $s_{\theta,t} \leftarrow s_{\theta,t}(\boldsymbol{y}_{s_t}, \{\boldsymbol{y}_{i,t}\}_{i=1}^K, \boldsymbol{y}_C)$ |
|   **end for** |   $\{s_{\phi_i,t}\}_{i=1}^K \leftarrow \{s_{\phi_i,t}(\boldsymbol{y}_{s_t}, \boldsymbol{y}_{i,t}, \boldsymbol{y}_C)\}_{i=1}^K$ |
|   $\theta, \{\phi_i\}_{i=i}^K \leftarrow \texttt{optim}(\frac{1}{n_B} \sum_{b=1}^{n_B} \mathcal{L}_b)$ |   $\boldsymbol{y}_{s_t} \leftarrow \boldsymbol{y}_{s_t} + \frac{\alpha}{2} s_{\theta,t} + \sqrt{\alpha} z_s; z_s \sim \mathcal{N}(0, I)$ |
| **end for** |   $\boldsymbol{y}_{i_t} \leftarrow \boldsymbol{y}_{i_t} + \frac{\alpha}{2} s_{\phi_i,t} + \sqrt{\alpha} z_i; z_i \sim \mathcal{N}(0, I)$ |
| | **end for** |

**Sampling** Given a trained conditional `Twigs` model, our generative process begins by sampling an external context or conditioning value $\mathbf{y}_C$, which can also be supplied externally. We then simulate the reverse diffusion process, similar to the one described in Equation 8, but with a modified score function to generate the data. The proposed algorithm for generating new data samples with `Twigs` is given in Algorithm 2 and involves a loop of updates between processes: the stem score network $s_{\phi_i}$ evolves the property $\mathbf{y}_i$, integrating information from the structure $\mathbf{y}_s$, and subsequently, the updated property information from $s_{\phi_i}$ is integrated into the main process by the score network $s_\theta$.

## 4 Experiments

We conduct a set of comprehensive experiments to demonstrate that `Twigs` improves over contemporary conditional generation methods. Benchmarks include: molecule generation conditioned over single (§ 4.1), and multiple (§ 4.2) properties on QM9, as well as molecule optimization on ZINC250K (§ 4.3), and network-graph generation conditioned on desired properties (§ 4.4).

Figure 2: First row: Samples by `Twigs` for 3D molecules conditioned on single properties on QM9. Second row: KDE and KL divergence results between target and predicted properties.

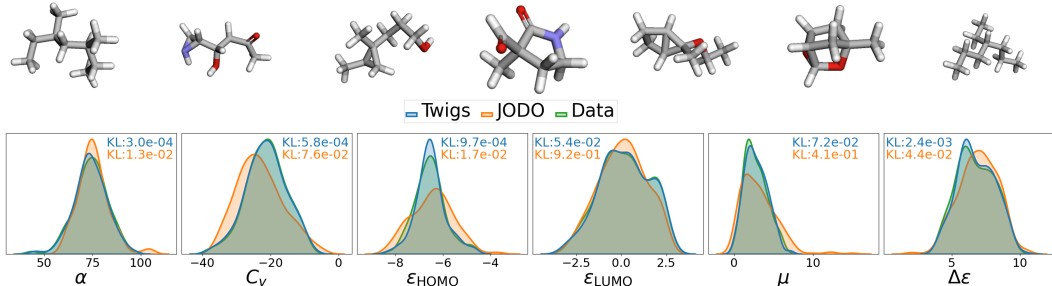

Table 3: MAE↓ results on single target quantum property for the QM9 dataset.

| Method | $C_v$ | $\mu$ | $\alpha$ | $\Delta\epsilon$ | $\epsilon_{HOMO}$ | $\epsilon_{LUMO}$ |
|---|---|---|---|---|---|---|
| EDM | 1.065 ($\pm$ 0.010) | 1.123 ($\pm$ 0.013) | 2.78 ($\pm$ 0.04) | 671 ($\pm$ 5) | 371 ($\pm$ 2) | 601 ($\pm$ 7) |
| GeoLDM | 1.025 ($\pm$ na) | 1.108 ($\pm$ na) | 2.37 ($\pm$ na) | 587 ($\pm$ na) | 340 ($\pm$ na) | 522 ($\pm$ na) |
| EEGSDE | 0.941 ($\pm$ 0.005) | 0.777 ($\pm$ 0.007) | 2.50 ($\pm$ 0.02) | 487 ($\pm$ 3) | 302 ($\pm$ 2) | 447 ($\pm$ 6) |
| EquiFM | 1.033 ($\pm$ na) | 1.106 ($\pm$ na) | 2.41 ($\pm$ na) | 591 ($\pm$ na) | 337 ($\pm$ na) | 530 ($\pm$ na) |
| TEDMol | 0.847 ($\pm$ na) | 0.840 ($\pm$ na) | 2.24 ($\pm$ na) | 443 ($\pm$ na) | 279 ($\pm$ na) | 412 ($\pm$ na) |
| JODO | 0.581 ($\pm$ 0.001) | 0.628 ($\pm$ 0.003) | 1.42 ($\pm$ 0.01) | 335 ($\pm$ 3) | 226 ($\pm$ 1) | 256 ($\pm$ 1) |
| `Twigs` | **0.559** ($\pm$ 0.002) | **0.627** ($\pm$ 0.001) | **1.36** ($\pm$ 0.01) | **323** ($\pm$ 2) | **225** ($\pm$ 1) | **244** ($\pm$ 3) |

Table 4: Novelty, atom & molecule stability for QM9 single property.

| | Novelty↑ | Atom Stability↑ | Mol Stability↑ | Novelty↑ | Atom Stability↑ | Mol Stability↑ |
|---|---|---|---|---|---|---|
| | $C_v$ | | | $\mu$ | | |
| EDM | 83.64($\pm$ 0.30) | 98.25($\pm$ 0.02) | 80.82($\pm$ 0.32) | 83.93($\pm$ 0.11) | 98.17($\pm$ 0.04) | 80.25($\pm$ 0.40) |
| EEGSDE | 83.53($\pm$ 0.18) | 98.25($\pm$ 0.06) | 80.83($\pm$ 0.33) | 83.85($\pm$ 0.20) | 98.18($\pm$ 0.02) | 80.25($\pm$ 0.18) |
| TEDMol | 83.82($\pm$ na) | 98.27($\pm$ na) | 80.83($\pm$ na) | 84.88($\pm$ na) | 98.22($\pm$ na) | 80.31($\pm$ na) |
| JODO | 91.21($\pm$ 0.22) | 97.74($\pm$ 0.29) | 91.75($\pm$ 0.11) | 91.22($\pm$ 0.02) | 99.02($\pm$ 0.02) | 92.86($\pm$ 0.15) |
| Twigs | **93.16**($\pm$ 0.16) | **99.14**($\pm$ 0.04) | **92.72**($\pm$ 0.07) | **92.90**($\pm$ 0.08) | **99.25**($\pm$ 0.05) | **93.91**($\pm$ 0.03) |
| | $\Delta\varepsilon$ | | | $\varepsilon_{HOMO}$ | | |
| EDM | 83.93($\pm$ 0.45) | 98.30($\pm$ 0.04) | 81.95($\pm$ 0.27) | 84.35($\pm$ 0.31) | 98.17($\pm$ 0.07) | 79.61($\pm$ 0.32) |
| EEGSDE | 84.09($\pm$ 0.27) | 98.18($\pm$ 0.06) | 80.99($\pm$ 0.29) | 84.44($\pm$ 0.33) | 98.19($\pm$ 0.03) | 79.81($\pm$ 0.20) |
| TEDMol | 84.92($\pm$ na) | 98.19($\pm$ na) | 79.82($\pm$ na) | 84.58($\pm$ na) | 98.22($\pm$ na) | 80.97($\pm$ na) |
| JODO | 91.02($\pm$ 0.19) | 98.42($\pm$ 0.02) | 93.32($\pm$ 0.04) | 91.38($\pm$ 0.02) | 98.19($\pm$ 0.38) | 92.02($\pm$ 0.03) |
| Twigs | **92.70**($\pm$ 0.04) | **99.31**($\pm$ 0.01) | **94.12**($\pm$ 0.31) | **93.02**($\pm$ 0.21) | **99.26**($\pm$ 0.04) | **94.11**($\pm$ 0.26) |
| | $\alpha$ | | | $\varepsilon_{LUMO}$ | | |
| EDM | 84.56($\pm$ 0.47) | 98.13($\pm$ 0.04) | 79.33($\pm$ 0.30) | 84.62($\pm$ 0.28) | 98.26($\pm$ 0.04) | 81.34($\pm$ 0.29) |
| EEGSDE | 84.19($\pm$ 0.32) | 98.26($\pm$ 0.03) | 80.95($\pm$ 0.35) | 84.83($\pm$ 0.30) | 98.14($\pm$ 0.01) | 80.00($\pm$ 0.21) |
| TEDMol | 85.82($\pm$ na) | 98.42($\pm$ na) | 82.03($\pm$ na) | 84.90($\pm$ na) | 98.31($\pm$ na) | 81.40($\pm$ na) |
| JODO | 90.15($\pm$ 0.02) | 98.74($\pm$ 0.05) | 94.03($\pm$ 0.32) | 90.78($\pm$ 0.42) | 98.84($\pm$ 0.04) | 94.02($\pm$ 0.03) |
| Twigs | **92.88**($\pm$ 0.13) | **99.28**($\pm$ 0.12) | **94.12**($\pm$ 0.02) | **92.48**($\pm$ 0.15) | **99.29**($\pm$ 0.17) | **94.11**($\pm$ 0.33) |

## 4.1 Single Quantum properties on QM9

**Setup.** We evaluate the effectiveness of `Twigs` for generating molecules with a single desired quantum property, sourced from the QM9 dataset [58], specifically, we consider $C_v$, $\mu$, $\alpha$, $\Delta\epsilon$, $\epsilon_{LUMO}$ and $\epsilon_{HOMO}$. To ensure consistency and comparability with the baselines, which include JODO [28], EDM [26], EEGSDE [3], GeoLDM [82], TEDMol [49], EquiFM [68], we adhere to the identical dataset preprocessing, training/test data partitions, and evaluation metrics outlined by Huang et al. [28]. Regarding parameterization of `Twigs`, we follow the attention architecture defined in Section B.1 with a single stem process.

**Results.** In Table 3, we report the Mean Absolute Error (MAE) results, and in Table 4, the Novelty, Atom stability and Molecule stability. Our method outperforms all the evaluated baselines across the specified properties. In Figure 2, the bottom row provides a Kernel Density Estimation (KDE) visualization which shows that `Twigs` achieves a more accurate distribution for the property values when compared with JODO, while the top row shows some 3D molecule samples by our model.

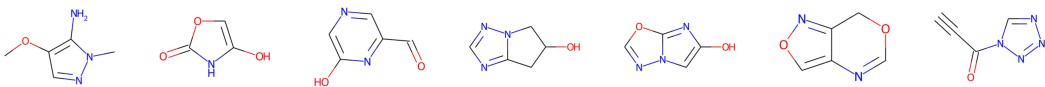

Figure 3: Samples of multiple-property conditional molecules by `Twigs` ($C_v$ and $\mu$) for QM9.

Table 5: MAE ($\downarrow$) for conditional generation on QM9 with multiple properties.

|         | $C_v$ | $\mu$ | $\Delta\epsilon$ | $\mu$ | $\alpha$ | $\mu$ |
|---------|-------|-------|-----------------|-------|----------|-------|
| EDM     | $1.097_{(\pm 0.007)}$ | $1.156_{(\pm 0.011)}$ | $683_{(\pm 1)}$ | $1.130_{(\pm 0.007)}$ | $2.76_{(\pm 0.01)}$ | $1.158_{(\pm 0.002)}$ |
| EEGSDE  | $0.981_{(\pm 0.008)}$ | $0.912_{(\pm 0.006)}$ | $563_{(\pm 3)}$ | $0.866_{(\pm 0.003)}$ | $2.61_{(\pm 0.01)}$ | $0.855_{(\pm 0.007)}$ |
| TEDMol  | $0.645_{(\pm \text{ n/a})}$ | $0.836_{(\pm \text{ n/a})}$ | $489_{(\pm \text{ n/a})}$ | $0.843_{(\pm \text{ n/a})}$ | $2.27_{(\pm \text{ n/a})}$ | $0.809_{(\pm \text{ n/a})}$ |
| JODO    | $0.634_{(\pm 0.002)}$ | $0.716_{(\pm 0.006)}$ | $350_{(\pm 4)}$ | $0.752_{(\pm 0.006)}$ | $1.52_{(\pm 0.01)}$ | $0.717_{(\pm 0.006)}$ |
| `Twigs` | $\mathbf{0.602}_{(\pm 0.001)}$ | $\mathbf{0.708}_{(\pm 0.002)}$ | $\mathbf{343}_{(\pm 2)}$ | $\mathbf{0.740}_{(\pm 0.003)}$ | $\mathbf{1.46}_{(\pm 0.01)}$ | $\mathbf{0.712}_{(\pm 0.002)}$ |

## 4.2 Multiple Quantum properties on QM9

**Setup.** This experiment evaluates the capability to combine multiple desired properties in the generated molecule. Specifically we follow Huang et al. [29] and consider all possible combinations of properties involving $\mu$: $(C_v, \mu)$, $(\Delta\epsilon, \mu)$, $(\alpha, \mu)$. Since we model two properties, we test our `Twigs` with two stem networks within the attention architecture described in Section B.1. We benchmark against several contemporary baselines, including EDM [26], EEGSDE [3] and JODO [28].

**Results.** In Table 5, we present the Mean Absolute Error (MAE) results obtained from the property predictors introduced by Huang et al. [28] for the various property pairs under consideration. The superior performance of `Twigs` across all baselines reinforces the findings from the single property experiment (Section 4.1), emphasizing the benefits of learning multiple hierarchical stem processes.

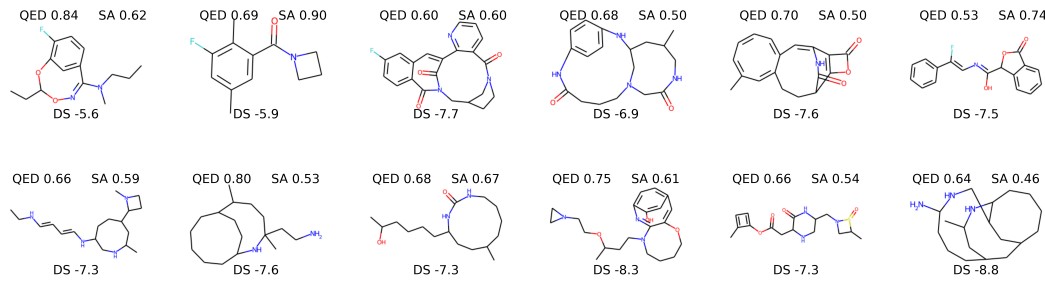

Figure 4: Molecules generated by `Twigs` from ZINC250k conditioned on fa7 (top), parp1 (bottom).

## 4.3 Molecule optimization on ZINC250K

**Setup.** The goal is to generate molecules from the ZINC250K dataset that exhibit optimal binding affinity, drug-likeness, and synthesizability for the following five target proteins: *parp1, fa7, 5ht1b, braf, jak2*. We adhere to the evaluation protocol established by Lee et al. [45], which involves generating 3000 molecules and assessing them using two metrics that constrain the desired properties, including docking score (DS), drug-likeness (QED), and synthetic accessibility (SA).

The first metric, *Novel hit ratio (%)*, represents the fraction of unique *hit molecules* that have a maximum Tanimoto similarity of less than 0.4 with the training molecules. Hit molecules are defined as those meeting the criteria: DS < (the median DS of the known active molecules), QED > 0.5, and SA < 5. The second metric, *Novel top 5% docking score*, is the average DS of the top 5% unique molecules that satisfy QED > 0.5 and SA < 5, with a maximum similarity of less than 0.4 to the training molecules.

**Baselines.** We consider REINVENT [55]: a reinforcement learning (RL) model that utilizes a prior sequence model, MORLD [33]: a RL model that uses QED and SA scores as intermediate rewards and docking scores as final rewards, HierVAE [34]: a VAE-based model that utilizes hierarchical molecular representation and active learning, GDSS [37]: a score-based diffusion model that evolves nodes and edge information with a system of SDEs, MOOD [45]: a score-based diffusion model based on GDSS that trains an additional property predictor to improve conditional generation. For MOOD we consider the version without the out-of-distribution (OOD) control, to have a fair comparison with our method. For `Twigs` we follow the GCN-based architecture described in Section B.2, with multiple stem processes (one for each target protein).

**Results.** In Table 6 we report the results for top 5% docking scores. We observe that `Twigs` achieves the highest score across all properties, excluding braf, where it achieves the second-best score after MOOD. In Table 7 we report the results for Novel hit ratio. The outcomes confirm that our model is improving the performance substantially over all the considered properties, except for braf, on which `Twigs` is the second-best performing model after MOOD. In Figure 4, we provide some samples of the molecules obtained by `Twigs` with the respective QED, SA, and docking score. Additionally, in Table 13 we report the MAE values for generating molecules with a desired target protein property, and in Table 14 we compare the inference cost of `Twigs` against MOOD.

Table 6: Novel top 5% docking score on ZINC250K. Best is **boldfaced**, second-best is in gray .

| Model | *parp1* | *fa7* | *5ht1b* | *braf* | *jak2* |
|---|---|---|---|---|---|
| REINVENT | 8.702$(\pm 0.523)$ | 7.205$(\pm 0.264)$ | 8.770$(\pm 0.316)$ | 8.392$(\pm 0.400)$ | 8.165$(\pm 0.277)$ |
| MORLD | 7.532$(\pm 0.260)$ | 6.263$(\pm 0.165)$ | 7.869$(\pm 0.650)$ | 8.040$(\pm 0.337)$ | 7.816$(\pm 0.133)$ |
| HierVAE | 9.487$(\pm 0.278)$ | 6.812$(\pm 0.274)$ | 8.081$(\pm 0.252)$ | 8.978$(\pm 0.525)$ | 8.285$(\pm 0.370)$ |
| GDSS | 9.967$(\pm 0.028)$ | 7.775$(\pm 0.039)$ | 9.459$(\pm 0.101)$ | 9.224$(\pm 0.068)$ | 8.926$(\pm 0.089)$ |
| MOOD | 10.409$(\pm 0.030)$ | 7.947$(\pm 0.034)$ | 10.487$(\pm 0.069)$ | **10.421**$(\pm 0.050)$ | 9.575$(\pm 0.075)$ |
| Twigs | **10.449**$(\pm 0.009)$ | **8.182**$(\pm 0.012)$ | **10.542**$(\pm 0.025)$ | 10.343$(\pm 0.024)$ | **9.678**$(\pm 0.032)$ |

Table 7: Novel hit ratio ($\uparrow$) results on ZINC250K.

| Model | *parp1* | *fa7* | *5ht1b* | *braf* | *jak2* |
|---|---|---|---|---|---|
| REINVENT | 0.480$(\pm 0.344)$ | 0.213$(\pm 0.081)$ | 2.453$(\pm 0.561)$ | 0.127$(\pm 0.088)$ | 0.613$(\pm 0.167)$ |
| MORLD | 0.047$(\pm 0.050)$ | 0.007$(\pm 0.013)$ | 0.880$(\pm 0.735)$ | 0.047$(\pm 0.040)$ | 0.227$(\pm 0.118)$ |
| HierVAE | 0.553$(\pm 0.214)$ | 0.007$(\pm 0.013)$ | 0.507$(\pm 0.278)$ | 0.207$(\pm 0.220)$ | 0.227$(\pm 0.127)$ |
| GDSS | 1.933$(\pm 0.208)$ | 0.368$(\pm 0.103)$ | 4.667$(\pm 0.306)$ | 0.167$(\pm 0.134)$ | 1.167$(\pm 0.281)$ |
| MOOD | 3.400$(\pm 0.117)$ | 0.433$(\pm 0.063)$ | 11.873$(\pm 0.521)$ | **2.207**$(\pm 0.165)$ | 3.953$(\pm 0.383)$ |
| Twigs | **3.733**$(\pm 0.081)$ | **0.900**$(\pm 0.012)$ | **16.366**$(\pm 0.029)$ | 1.933$(\pm 0.023)$ | **5.100**$(\pm 0.312)$ |

### 4.4 Generation of Network graphs with desired properties

**Setup.** We follow the data processing delineated by Jo et al. [37] and provide results for the Community-small [60] and Enzymes datasets [62]. To test the capabilities to generate conditional graphs, we extract four properties via the NetworkX library [20], including density, clustering, assortativity, and transitivity. Considering a graph $G$ with $n$ nodes and $m$ edges, we have: (1) Density: $d = \frac{2m}{n(n-1)}$, (2) Clustering coefficient: the average $C = \frac{1}{n} \sum_{v \in G} c_v$. (3) Assortativity: measures the similarity of connections in the graph with respect to the node degree. (4) Transitivity: the fraction of all possible triangles present in $G$. Possible triangles are identified by the number of "triads" (two edges with a shared vertex). The transitivity is $T = 3\frac{\#triangles}{\#triads}$.

**Baselines.** In terms of baselines, we first consider two versions of MOOD [45] (two OOD coefficients), and we train the property predictors using the codes from the authors. Our second baseline is GDSS [37], which we modify to be equipped with a classifier-free guidance scheme. We also consider the version of GDSS based on transformers, which leverages the graph-multi-head attention [2]. Finally, we consider Digress [75], which is a classifier-based guidance diffusion model based on attention mechanisms. We parameterize our `Twigs` model with our GCN architecture described in Section B.2, with a single stem process.

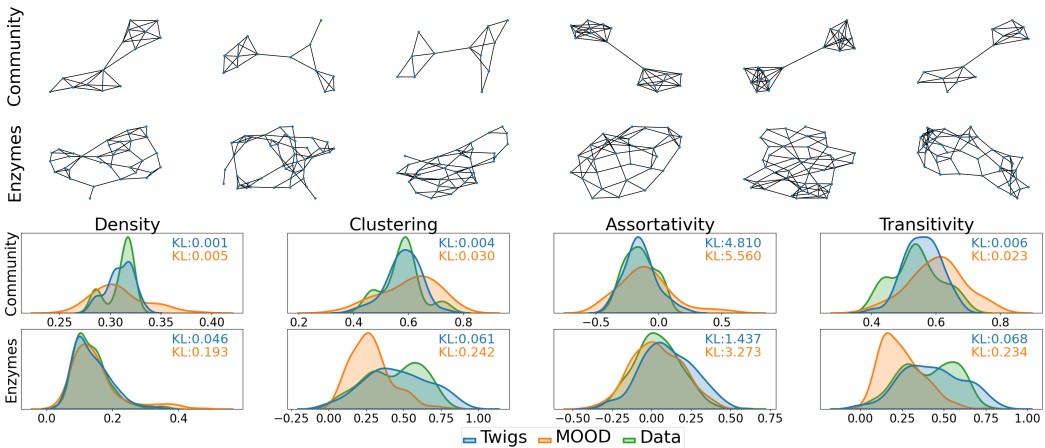

Figure 5: Visualization of Community-small and Enzymes datasets. First and second rows: samples generated by `Twigs`. Third and fourth rows: KDE plots and corresponding KL divergence values.

Table 8: MAE (↓) values on Community-small and Enzymes, conditioned on single properties.

| Model | Community Small | | | | Enzymes | | | |
|---|---|---|---|---|---|---|---|---|
| | Density | Clustering | Assortativity | Transitivity | Density | Clustering | Assortativity | Transitivity |
| GDSS | 2.95 | 12.1 | 19.6 | 11.4 | 8.04 | 2.53 | 1.98 | 2.55 |
| GDSS-T | 2.30 | 11.5 | 19.2 | 10.1 | 9.25 | 3.27 | 2.03 | 2.68 |
| Digress | 2.34 | 10.6 | 17.8 | 9.42 | 8.04 | 2.39 | 1.95 | 2.55 |
| MOOD-1 | 2.35 | 11.1 | 18.8 | 10.5 | 7.94 | 2.34 | 1.83 | 2.12 |
| MOOD-4 | 2.12 | 11.3 | 16.7 | 8.76 | 7.98 | 2.44 | 1.99 | 2.43 |
| Twigs | **2.07** | **9.67** | **15.2** | **8.35** | **7.35** | **2.23** | **1.72** | **2.03** |

**Results.** Table 8 reports the MAE average of three runs, demonstrating that `Twigs` consistently outperforms the considered baselines on all cases across the two datasets. MOOD is the second-best performing model in the majority of the cases. We further strengthen the MAE results by providing in Figure 5 (bottom) the KDE plots of the property distributions of the graph generated by `Twigs` and MOOD. The Figure demonstrates that `Twigs` can achieve a higher fidelity to the data, which is also confirmed by the lower KL divergence values. Figure 5 (top) depicts some random graph samples generated by `Twigs`.

### 4.5 Ablation study on multiple properties

**Setup.** Assuming conditional independence among the properties $\alpha$, $\epsilon_{\text{HOMO}}$, $\epsilon_{\text{LUMO}}$, $\Delta\epsilon$, $\mu$, and $C_v$ given the molecular graph can simplify the modeling process. This assumption leverages the fact that the molecular graph captures the essential structural dependencies, allowing us to treat the properties as independent for computational efficiency and ease of interpretation, even if slight interdependencies exist.

**Results.** Here we show that such modeling assumption can work practically. Table 9 reports the MAE on molecular graphs for QM9 on three properties, showing that our method consistently achieves lower error on all the properties. Table 10 shows that on generic graphs `Twigs` can achieve lower MAE on all the considered cases, in the cases of two and three properties.

Table 9: MAE values over three properties for QM9.

| Model | $\alpha$ | $\mu$ | $\Delta\epsilon$ |
|---|---|---|---|
| JODO | 2.749 (± 0.03) | 1.162 (± 0.04) | 717 (± 5) |
| Twigs | **2.544** (± 0.05) | **1.094** (± 0.02) | **640** (± 3) |

Table 10: MAE results for two and three properties on community small.

| Model | Pair1 | | Pair2 | | Triplet | | |
|---|---|---|---|---|---|---|---|
| | Density | Clustering | Density | Assortativity | Density | Clustering | Assortativity |
| GDSS | 2.95 | 13.3 | 2.61 | 19.8 | 2.97 | 12.5 | 19.4 |
| Digress | 2.82 | 12.1 | 2.52 | 18.1 | 2.65 | 11.2 | 18.2 |
| MOOD | 2.43 | 12.0 | 2.40 | 17.2 | 2.53 | 11.4 | 17.3 |
| Twigs | **2.34** | **11.0** | **2.39** | **16.7** | **2.27** | **10.6** | **16.1** |

## 4.6 Training time

In Table 11 we study the impact of multiple diffusion flows on the community-small and Enzymes datasets. Specifically, we report the average time for the overall training for `Twigs` with one and three secondary diffusion flows. We observe that our models encounter a small overhead compared to GDSS and Digress, however, we believe it is a good tradeoff because it achieves a lower MAE.

Table 11: Overall training time for 5,000 epochs (hours and minutes) for `Twigs` with different secondary diffusion flows, GDSS, and Digress on the Community-small and Enzymes datasets.

| Dataset | Twigs $p = 1$ | Twigs $p = 3$ | GDSS | Digress |
|---|---|---|---|---|
| Community-small | 0h 22m | 0h 24m | 0h 19m | 0h 20m |
| Enzymes | 6h 45m | 6h 59m | 6h 42m | 6h 43m |

## 5 Conclusion, Broader Implications, and Limitations

We introduced a novel approach to model conditional information within generative models tailored for graph data. `Twigs` incorporates the novel mechanism of *loop guidance* to control the overall generative process by first bifurcating the diffusion flow into multiple stem processes and then re-integrating them into the trunk process, resembling a loop. Our experimental results showcase the performance gains of `Twigs` when compared to current state-of-the-art baselines across various conditional graph generation tasks.

Conditional generation is fast emerging as one of the most exciting avenues within machine learning and would benefit from techniques beyond classifier-based and classifier-free schemes, making our method applicable to settings beyond this work. Indeed, while the current work has focused on graph settings, `Twigs` might find use in other domains (e.g., image, text, and audio). However, whether `Twigs` is effective in such settings needs to be investigated in future works.

Training multiple properties (stem processes) might require training additional parameters, incurring additional computation and training time. Our ablation study on training time due to multiple processes (Section 4.6) suggests that `Twigs` could provide a good tradeoff (lower MAE compared to some prominent existing methods at the expense of small additional computational overhead).

Finally, assuming factorization of the distribution over stem processes conditioned on the trunk process might not always be realistic. Our experiments in Section 4.5 suggest that `Twigs` might still be able to achieve a strong performance when considering multiple properties. In case some prior knowledge is available about some properties that violate this assumption, we could, in principle, adapt `Twigs` by grouping them into a single stem process while factorizing with the remaining ones.

## Acknowledgments

YV and VG acknowledge support from the Research Council of Finland for the "Human-steered next-generation machine learning for reviving drug design" project (grant decision 342077). VG also acknowledges Jane and Aatos Erkko Foundation (grant 7001703) for "Biodesign: Use of artificial intelligence in enzyme design for synthetic biology". GM acknowledges support from the Engineering and Physical Sciences Research Council (EPSRC) and the BBC under iCASE. AF is partially funded by the CRUK National Biomarker Centre, by the Manchester Experimental Cancer Medicine Centre and the NIHR Manchester Biomedical Research Centre.

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

# A Proofs

## A.1 Derivation of the reverse SDE

For a Stochastic Differential Equation (SDE) of the form,

$$dx = f(x_t, t)\mathrm{d}t + g(x_t, t)\mathrm{d}\mathbf{w} \tag{10}$$

where $f(\cdot)$ and $g(\cdot)$ are diffusion, drift function and $\mathrm{d}\mathbf{w}$ is the weiner noise. The evolution of the distribution of $x_t$ is governed by the Kolmogorov Forward Equation (KFE) as,

$$\partial_t p(x_t) = -\partial_{x_t}\left[f(x_t) p(x_t)\right] + \frac{1}{2}\partial_{x_t}^2\left[g^2(x_t) p(x_t)\right] \tag{11}$$

**Kolmogrov Forward/Backward Equation (KFE/KBE).** Essentially KFE describes the evolution of a probability distribution $p(x_t)$ forward in time. The reverse-time SDE can be derived by solving the Kolmogorov Backward Equation (K.B.E) as derived in Anderson [1]. It can be defined for $t_1 \geq t_0$ as,

$$-\partial_t p(x_{t_1} \mid x_{t_0}) = f(x_{t_0})\partial_{x_{t_0}} p(x_{t_1} \mid x_{t_0}) + \frac{1}{2}g^2(x_{t_0})\partial_{x_{t_0}}^2 p(x_{t_1} \mid x_{t_0}) \tag{12}$$

where $x_{t_0}$ and $x_{t_1}$ are distributions at the respective time steps. Specifically, it models how the distribution dynamics at a later point $t_1$ in time changes as we change $t_0$ at an earlier time.

In our case, we consider the diffusion over structure $\mathbf{y}_s$ and properties $\{\mathbf{y}_1, \ldots, \mathbf{y}_k\}$. The KFE of the system $\mathbf{y} = \{\mathbf{y}_s, \mathbf{y}_1, \ldots, \mathbf{y}_k\}$ is given by,

$$\partial_t p(\mathbf{y}_t) = -\partial_{\mathbf{y}_t}\left[f(\mathbf{y}_t) p(\mathbf{y}_t)\right] + \frac{1}{2}\partial_{\mathbf{y}_t}^2\left[g^2(\mathbf{y}_t) p(\mathbf{y}_t)\right] \tag{13}$$

**Independence Factorization.** We can factorize $p(\mathbf{y}_t)$ based on our assumption that the properties $\{\mathbf{y}_{1,t}, \ldots, \mathbf{y}_{k,t}\}$ are independent conditioned on the structure $\mathbf{y}_{s,t}$ as

$$\begin{aligned} p(\mathbf{y}_t) &= p(\mathbf{y}_{s,t}, \mathbf{y}_{1,t}, \ldots, \mathbf{y}_{k,t}) \\ &= p(\mathbf{y}_{s,t}) p(\mathbf{y}_{1,t}, \ldots, \mathbf{y}_{k,t} \mid \mathbf{y}_{s,t}) \\ &= p(\mathbf{y}_{s,t}) \prod_i^k p(\mathbf{y}_{i,t} \mid \mathbf{y}_{s,t}) \end{aligned} \tag{14}$$

Leveraging this factorization, we can define a system of SDEs with KFEs for each variable, leading us to the SDE system defined in Eq. 1 and Eq. 2.

**Reverse SDE:** In the reverse case, we aim to denoise the full vector $\mathbf{y} = \{\mathbf{y}_s, \mathbf{y}_1, \ldots, \mathbf{y}_k\}$ where $\mathbf{y}_s$ denotes the diffusion over structure and $\{\mathbf{y}_1, \ldots, \mathbf{y}_k\}$ over the $k$ properties via reverse SDE. Expressing in the form of Eq. 12, we note that for $t_1 \geq t_0$,

$$-\partial_t p(\mathbf{y}_{t_1} \mid \mathbf{y}_{t_0}) = f(\mathbf{y}_{t_0})\partial_{\mathbf{y}_{t_0}} p(\mathbf{y}_{t_1} \mid \mathbf{y}_{t_0}) + \frac{1}{2}g^2(\mathbf{y}_{t_0})\partial_{\mathbf{y}_{t_0}}^2 p(\mathbf{y}_{t_1} \mid \mathbf{y}_{t_0}) \tag{15}$$

Anderson [1] defines a joint distribution over the time-ordered variables $\mathbf{y}_{t_1}$ and $\mathbf{y}_{t_0}$ to derive the reverse SDE. We utilize their analysis and define a joint distribution

$$\begin{aligned} p(\mathbf{y}_{t_1}, \mathbf{y}_{t_0}) &:= p(\mathbf{y}_{s,t_1}, \mathbf{y}_{1,t_1}, \ldots, \mathbf{y}_{k,t_1}, \mathbf{y}_{s,t_0}, \mathbf{y}_{1,t_0}, \ldots, \mathbf{y}_{k,t_0}) \\ &= p(\mathbf{y}_{s,t_1}, \mathbf{y}_{1,t_1}, \ldots, \mathbf{y}_{k,t_1} \mid \mathbf{y}_{s,t_0}, \mathbf{y}_{1,t_0}, \ldots, \mathbf{y}_{k,t_0}) p(\mathbf{y}_{s,t_0}, \mathbf{y}_{1,t_0}, \ldots, \mathbf{y}_{k,t_0}) \end{aligned} \tag{16}$$

We denote $p(\mathbf{y}_{s,t_0}, \mathbf{y}_{1,t_0}, \ldots, \mathbf{y}_{k,t_0})$ by $p(\mathbf{y}_{t_0})$, and note that it can be decomposed similarly as in Eq. 14. Taking the time derivative of Eq. 16, we get

$$\begin{aligned} -\partial_t p(\mathbf{y}_{t_1}, \mathbf{y}_{t_0}) = &-\partial_t p(\mathbf{y}_{s,t_1}, \mathbf{y}_{1,t_1}, \ldots, \mathbf{y}_{k,t_1} \mid \mathbf{y}_{s,t_0}, \mathbf{y}_{1,t_0}, \ldots, \mathbf{y}_{k,t_0}) p(\mathbf{y}_{t_0}) \\ &- \partial_t p(\mathbf{y}_{t_0}) p(\mathbf{y}_{s,t_1}, \mathbf{y}_{1,t_1}, \ldots, \mathbf{y}_{k,t_1} \mid \mathbf{y}_{s,t_0}, \mathbf{y}_{1,t_0}, \ldots, \mathbf{y}_{k,t_0}) \end{aligned} \tag{17}$$

**Comparison with KFE/KBE.** We observe that $\partial_t p(\mathbf{y}_{s,t_1}, \mathbf{y}_{1,t_1}, \ldots, \mathbf{y}_{k,t_1} \mid \mathbf{y}_{s,t_0}, \mathbf{y}_{1,t_0}, \ldots, \mathbf{y}_{k,t_0})$ corresponds to the KBE in Eq. 15 and $\partial_t p(\mathbf{y}_{t_0})$ to the KFE in Eq. 13. Denoting

$\{\mathbf{y}_{s,t_1}, \mathbf{y}_{1,t_1}, \ldots, \mathbf{y}_{k_t1}\}$ by $\mathbf{y}_{t_1}$, we immediately get

$$- \partial_t p\left(\mathbf{y}_{t_1} \mid \mathbf{y}_{t_0}\right) p(\mathbf{y}_{t_0}) - \partial_t p(\mathbf{y}_{t_0}) p\left(\mathbf{y}_{t_1} \mid \mathbf{y}_{t_0}\right)$$
$$= \left( f\left(\mathbf{y}_{t_0}\right) \partial_{\mathbf{y}_{t_0}} p\left(\mathbf{y}_{t_1} \mid \mathbf{y}_{t_0}\right) + \frac{1}{2} g^2\left(\mathbf{y}_{t_0}\right) \partial^2_{\mathbf{y}_{t_0}} p\left(\mathbf{y}_{t_1} \mid \mathbf{y}_{t_0}\right) \right) p(\mathbf{y}_{t_0}) \tag{18}$$
$$+ p\left(\mathbf{y}_{t_1} \mid \mathbf{y}_{t_0}\right) \left( \partial_{\mathbf{y}_{t_0}} \left[ f\left(\mathbf{y}_{t_0}\right) p\left(\mathbf{y}_{t_0}\right) \right] - \frac{1}{2} \partial^2_{\mathbf{y}_{t_0}} \left[ g^2\left(\mathbf{y}_{t_0}\right) p\left(\mathbf{y}_{t_0}\right) \right] \right)$$

The derivatives can be handled, by following standard differentiation rules as,

$$\partial_{\mathbf{y}_{t_0}} p\left(\mathbf{y}_{t_1} \mid \mathbf{y}_{t_0}\right) = \partial_{\mathbf{y}_{t_0}} \left[ \frac{p\left(\mathbf{y}_{t_1}, \mathbf{y}_{t_0}\right)}{p\left(\mathbf{y}_{t_0}\right)} \right]$$
$$= \frac{\partial_{\mathbf{y}_{t_0}} p\left(\mathbf{y}_{t_1}, \mathbf{y}_{t_0}\right)}{p\left(\mathbf{y}_{t_0}\right)} - \frac{p\left(\mathbf{y}_{t_1}, \mathbf{y}_{t_0}\right) \partial_{\mathbf{y}_{t_0}} p\left(\mathbf{y}_{t_0}\right)}{p^2\left(\mathbf{y}_{t_0}\right)} \tag{19}$$

Evaluating the derivative of the products in the forward Kolmogorov equation and substituting the derivatives accordingly we obtain,

$$-\partial_t p\left(\mathbf{y}_{t_1}, \mathbf{y}_{t_0}\right) = \partial_{\mathbf{y}_{t_0}} \left[ f\left(\mathbf{y}_{t_0}\right) p\left(\mathbf{y}_{t_0}, \mathbf{y}_{t_1}\right) \right] + \frac{1}{2} g^2\left(\mathbf{y}_{t_0}\right) \partial^2_{\mathbf{y}_{t_0}} p\left(\mathbf{y}_{t_1} \mid \mathbf{y}_{t_0}\right) p(\mathbf{y}_{t_0})$$
$$- \frac{1}{2} p\left(\mathbf{y}_{t_1} \mid \mathbf{y}_{t_0}\right) \partial^2_{\mathbf{y}_{t_0}} \left[ g^2\left(\mathbf{y}_{t_0}\right) p(\mathbf{y}_{t_0}) \right] \tag{20}$$

Matching the terms of the second-order derivatives with the expansion of the derivative and doing some algebraic manipulations, we obtain

$$-\partial_t p\left(\mathbf{y}_{t_1}, \mathbf{y}_{t_0}\right) = \partial_{\mathbf{y}_{t_0}} \left[ f\left(\mathbf{y}_{t_0}\right) p\left(\mathbf{y}_{t_0}, \mathbf{y}_{t_1}\right) \right] + \frac{1}{2} \partial^2_{\mathbf{y}_{t_0}} \left[ p\left(\mathbf{y}_{t_1}, \mathbf{y}_{t_0}\right) g^2\left(\mathbf{y}_{t_0}\right) \right]$$
$$- \partial_{\mathbf{y}_{t_0}} \left[ p\left(\mathbf{y}_{t_1} \mid \mathbf{y}_{t_0}\right) \partial_{\mathbf{y}_{t_0}} \left[ g^2\left(\mathbf{y}_{t_0}\right) p\left(\mathbf{y}_{t_0}\right) \right] \right] , \tag{21}$$

which can be written as

$$-\partial_t p\left(\mathbf{y}_{t_1}, \mathbf{y}_{t_0}\right) = - \partial_{\mathbf{y}_{t_0}} \left[ p\left(\mathbf{y}_{t_1}, \mathbf{y}_{t_0}\right) \left( -f\left(\mathbf{y}_{t_0}\right) + \frac{1}{p\left(\mathbf{y}_{t_0}\right)} \partial_{\mathbf{y}_{t_0}} \left( g^2\left(\mathbf{y}_{t_0}\right) p\left(\mathbf{y}_{t_0}\right) \right) \right) \right] + \tag{22}$$
$$\frac{1}{2} \partial^2_{\mathbf{y}_{t_0}} \left[ p\left(\mathbf{y}_{t_1}, \mathbf{y}_{t_0}\right) g^2\left(\mathbf{y}_{t_0}\right) \right] \tag{23}$$

**Comparison with KFE.** The above result is in the form of a Kolmogorov forward equation with the joint probability distribution $p\left(\mathbf{y}_{t_1}, \mathbf{y}_{t_0}\right)$. The time-ordering is $t_1 > t_0$ and the term $-\partial_t p\left(\mathbf{y}_{t_1}, \mathbf{y}_{t_0}\right)$ describes the change of probability distribution as we move backward in time. We can marginalize over $t_1$, using the Leibniz rule, to obtain

$$-\partial_t p\left(\mathbf{y}_{t_0}\right) = -\partial_{\mathbf{y}_{t_0}} \left[ p\left(\mathbf{y}_{t_0}\right) \left( -f\left(\mathbf{y}_{t_0}\right) + \frac{1}{p\left(\mathbf{y}_{t_0}\right)} \partial_{\mathbf{y}_{t_0}} \left( g^2\left(\mathbf{y}_{t_0}\right) p\left(\mathbf{y}_{t_0}\right) \right) \right) \right] + \frac{1}{2} \partial^2_{\mathbf{y}_{t_0}} \left[ p\left(\mathbf{y}_{t_0}\right) g^2\left(\mathbf{y}_{t_0}\right) \right] \tag{24}$$

This finally gives a stochastic differential equation analogous to the Fokker-Planck/forward Kolmogorov equation that can be solved backward in time:

$$d\mathbf{y}_{t_0} = \left( -f(\mathbf{y}_{t_0}, t) + \frac{1}{p\left(\mathbf{y}_{t_0}\right)} \partial_{\mathbf{y}_{t_0}} \left( g^2\left(\mathbf{y}_{t_0}\right) p\left(\mathbf{y}_{t_0}\right) \right) \right) dt + g\left(\mathbf{y}_{t_0}\right) d\mathbf{w} \tag{25}$$

We keep $g^2\left(\mathbf{y}_{t_0}\right)$ independent of $\mathbf{y}_{t_0}$. Applying the log-derivative trick, the SDE simplifies to

$$d\mathbf{y}_{t_0} = (f(\mathbf{y}_{t_0}, t) - g^2_{t_0} \nabla_{\mathbf{y}_{t_0}} \log p(\mathbf{y}_{t_0})) dt + g_{t_0} d\mathbf{w} \tag{26}$$

## A.2 Conditional score factorization

We extend our method to incorporate an external context or conditional information for conditional generation, similar to classifier-based [8] and classifier-free [23] guidance. Following similar notation, the reverse SDE [67], given an external context $\mathbf{y}_C$ can be written as

$$d\mathbf{y}_t = [\mathbf{f}(\mathbf{y}_t, t) - g^2_t \nabla_{\mathbf{y}_t} \log p_t(\mathbf{y}_t, \mathbf{y}_C)] dt + g_t d\bar{\mathbf{w}} \tag{27}$$

Here $\mathbf{y}_t = \{\mathbf{y}_{s,t}, \mathbf{y}_{1,t}, \ldots, \mathbf{y}_{k,t}\}$, and $\mathbf{y}_C = \{\mathbf{y}_c \mid c \in C\}$ is an external context or conditioning variable. This external context can be a scalar or vector describing a property value of the primary variable like QED or plogp in the case of molecules or image labels in the case of images. The $\nabla_{\mathbf{y}_t} \log p_t(\mathbf{y}_t, \mathbf{y}_C)$ term pertains to the score function which guides the process (see table 2 for comparison with both classifier-based and classifier-free guidance). Under our condition independence assumption, the score function factorizes as

$$p_t(\mathbf{y}_{s,t}, \mathbf{y}_{1,t}, \ldots, \mathbf{y}_{k,t}, \mathbf{y}_C) = \prod_i^k p_t(\mathbf{y}_{i,t} \mid \mathbf{y}_{s,t}, \mathbf{y}_c) p_t(\mathbf{y}_{s,t}, \mathbf{y}_C) \tag{28}$$

$$\nabla_{\mathbf{y}_t} \log p_t(\mathbf{y}_{s,t}, \mathbf{y}_{1,t}, \ldots, \mathbf{y}_{k,t}, \mathbf{y}_C) = \nabla_{\mathbf{y}_t} \log p_t(\mathbf{y}_{s,t}, \mathbf{y}_C) + \sum_{i \notin C}^{k} \nabla_{\mathbf{y}_t} \log p_t(\mathbf{y}_{i,t} \mid \mathbf{y}_{s,t})$$
$$+ \sum_c^C \sum_i^k \delta_{i=c} \nabla_{\mathbf{y}_t} \log p_t(\mathbf{y}_{i,t} \mid \mathbf{y}_{s,t}, \mathbf{y}_c) \tag{29}$$

## B Parameterizations

Here we describe two instances of `Twigs` based on architecture choices: Attention networks, and graph convolution networks (GCNs). `Twigs` with attention is used in 4.1 and 4.2, while `Twigs` with GCNs is used in 4.3 and 4.4.

### B.1 Twigs with graph attention

We denote the variable $\mathbf{y}_s$ as a 3D graph $G = (A, x, h)$, with node coordinates $x = (x^1, \ldots, x^N) \in \mathbb{R}^{N \times 3}$, node features $h = (h^1, \ldots, h^N) \in \mathbb{R}^{N \times d1}$, and edge information $A \in \mathbb{R}^{N \times N \times d2}$. The variable $\mathbf{C} \in \mathbb{R}$ denotes the conditional information, which is obtained by adding the noise level $\log(\alpha_t^2/\sigma_t^2)$, the perturbed property $\mathbf{y}_i \sim \mathcal{N}(0, I) \in \mathbb{R}$, and the fixed property $\mathbf{y}_C \in \mathbb{R}$. The context $\mathbf{C}$ is combined with $\mathbf{y}_s$ by multilayer perceptions (MLP), after projecting $(h, A, x)$ respectively into $\mathbf{H}, \mathbf{E}, \mathbf{P}$:

$$\text{AdaLN} = (1 + \text{MLP}(\mathbf{C})) \cdot \text{LN}(\mathbf{H}) + \text{MLP}(\mathbf{C}) \tag{30}$$
$$\mathbf{M}^l = \text{MHA}(\text{AdaLN}(\mathbf{H}, \mathbf{C}), \text{AdaLN}(\mathbf{E}, \mathbf{C}), \mathbf{P})$$

where MHA is the multi-head attention, and AdaLN is Adaptive LayerNorm (LN) function. Subsequently, we leverage the Scale function $\text{Scale}(h, \mathbf{C}) = \text{MLP}(\mathbf{C}) \cdot h$, and the Feed Forward Network (FFN) to obtain the Diffusion Graph Transformer (DGT) block, as defined in [28], which is described by Eq (31)(32). DGT first computes the intermediate representations for the $l$-th layer as:

$$\mathbf{M}^l = \text{MHA}(\text{AdaLN}(\mathbf{H}^l, \mathbf{C}), \text{AdaLN}(\mathbf{E}^l, \mathbf{C}), \mathbf{P}^l) \tag{31}$$
$$\hat{\mathbf{H}} = \text{Scale}(\mathbf{M}^l, \mathbf{C}) + \mathbf{H}^l$$
$$\hat{\mathbf{E}} = \text{Scale}(\mathbf{M}_i^l + \mathbf{M}_j^l, \mathbf{C}) + \mathbf{E}^l$$

then computes the $l + 1$ layer as:

$$\mathbf{E}^{l+1} = \text{Scale}(\text{FFN}(\text{AdaLN}(\text{Scale}(\hat{\mathbf{E}}, C)), C) + \hat{\mathbf{E}} \tag{32}$$
$$\mathbf{H}^{l+1} = \text{Scale}(\text{FFN}(\text{AdaLN}(\hat{\mathbf{H}}, \mathbf{C})), \mathbf{C}) + \hat{\mathbf{H}}$$
$$\mathbf{P}_i^{l+1} = \sum_{i \neq j} \frac{\mathbf{P}_i^l - \mathbf{P}_j^l}{||\mathbf{P}_i^l - \mathbf{P}_j^l||^2} \tanh(\text{MLP}(\mathbf{E}_{i,j}^{l+1}))$$

The `Twigs` trunk process $s_\theta$ is parameterized as:

$$s_\theta = \text{DGT}(\mathbf{y}_s, \mathbf{y}_i, \mathbf{y}_C) + \sum_i \text{PDGT}_i(\mathbf{y}_s, \mathbf{y}_i, \mathbf{y}_C) \tag{33}$$

where $\text{PDGT}_i$ resembles the stem process networks $s_{\phi_i}$, which is obtained by pooling to a one-dimensional variable by an MLP operation, over the output of the DGT block. To optimize Eq (9), DGT minimizes the denoising score matching objective from [28] for node, edge and position information $(h, A, x)$, while $\text{PDGT}_i$ for the perturbed property $\mathbf{y}_i$.

## B.2 Twigs with graph convolutions

In the case of 2D graphs with $N$ nodes we consider the variable $\mathbf{y}_s = (\boldsymbol{X}, \boldsymbol{A}) \in \mathbb{R}^{N \times F} \times \mathbb{R}^{N \times N}$, where $F$ is the dimension of the node features, $\boldsymbol{X} \in \mathbb{R}^{N \times F}$ are node features, $\boldsymbol{A} \in \mathbb{R}^{N \times N}$ is weighted adjacency matrix. We define the perturbed property $\mathbf{y}_i \in \mathbb{R}$ and the (fixed) property $\mathbf{y}_C \in \mathbb{R}$. The stem process network $s_{\phi_i}$ is given as:

$$s_{\phi_i} = \mathrm{MLP}_i(\mathrm{GNN}(\mathrm{P}_i, \boldsymbol{A})); \quad \mathrm{P}_i = (\boldsymbol{X} \,\|\, \mathbf{v}_i, \| \mathbf{v}_C) \tag{34}$$

where $\mathbf{v}_i$ and $\mathbf{v}_C$ are vectors obtained by repeating $N$ times the perturbed property $\mathbf{y}_i$ and the fixed property $\mathbf{y}_C$ respectively, and concatenating them into the node features matrix $\boldsymbol{X}$. The `Twigs` trunk process $s_\theta$ is obtained by combining the contributions from the properties $\mathbf{y}_i$ derived by the stem processes $s_{\phi_i}$ and the structure $\mathbf{y}_s$, as follows:

$$s_\theta = s_{\theta X}(\boldsymbol{X}, \boldsymbol{A}, \mathbf{y}_C) + \sum_i s_{\phi_i}(\boldsymbol{X}, \mathbf{y}_i, \mathbf{y}_C) \tag{35}$$

where $s_{\theta X}$ is a conditional node feature score network: $s_{\theta X} = \mathrm{MLP}(\mathrm{GNN}(\boldsymbol{X} \,\|\, \mathbf{y}_C, \boldsymbol{A}))$. Finally, following [37], $\boldsymbol{A}$ is co-evolved together with the node features, by the adjacency score model $s_\theta^A$

$$\boldsymbol{s}_{\theta^A} = \mathrm{MLP}\left(\left[\{\mathrm{GMH}\left(\boldsymbol{H}_i, \boldsymbol{A}_t^p\right)\}_{i=0,p=1}^{K,P}\right]\right) \tag{36}$$

where GMH is graph multi-head attention [2], which employs higher-order adjacency matrices $\boldsymbol{A}_t^p$, and $K$ denotes the number of GMH layers. The optimization for the `Twigs` objective function (9), is obtained by minimizing the denoising score matching for $\boldsymbol{A}, \boldsymbol{X}, \mathrm{P}_i$.

The GMH block employs higher-order adjacency matrices $\boldsymbol{A}_t^p$ to represent the long-range dependencies and is provided as: $\boldsymbol{s}_{\theta^A}(\boldsymbol{G}_t) = \mathrm{MLP}\left(\left[\{\mathrm{GMH}\left(\boldsymbol{H}_i, \boldsymbol{A}_t^p\right)\}_{i=0,p=1}^{K,P}\right]\right)$.

# C Additional experimental results

## C.1 QM9 dataset

Further details for generation conditioned on quantum properties from Section 4.1.

**Molecular quality.** Additional results for molecular stability in 2D and Fréchet ChemNet Distance (FCD) for 2D and 3D are given in Table 12.

Table 12: Molecule quality results.

| Property | Mol-S-2D $\uparrow$ | FCD-2D $\downarrow$ | FCD-3D $\downarrow$ |
|---|---|---|---|
| $C_v$ | 98.88 | 0.107 | 0.871 |
| $\mu$ | 98.93 | 0.125 | 0.842 |
| $\alpha$ | 98.71 | 0.106 | 0.867 |
| $\Delta\epsilon$ | 98.82 | 0.105 | 0.787 |
| $\epsilon_{\mathrm{HOMO}}$ | 98.95 | 0.111 | 0.827 |
| $\epsilon_{\mathrm{LUMO}}$ | 98.52 | 0.117 | 0.846 |

## C.2 ZINC250K dataset

**Conditional generation.** The evaluation is performed by measuring the MAE of the pre-trained predictors released from [45], which given a molecule $G_t$ are trained to predict

$$\mathrm{Obj} = \widehat{\mathrm{DS}}(\boldsymbol{G}_t) \times \mathrm{QED}(\boldsymbol{G}_t) \times \widehat{\mathrm{SA}}(\boldsymbol{G}_t) \tag{37}$$

where $\widehat{\mathrm{DS}}$ is the normalized docking score (DS) of the considered target protein, QED is the drug-likeness, and $\widehat{\mathrm{SA}}$ is the normalized synthetic accessibility (SA).

In terms of baselines, we consider the MOOD model [45], which leverages a classifier-based guidance scheme, and we also implement a diffusion guidance version of GDSS [37] based on the classifier-free scheme. Our `Twigs` method is parameterized by the architecture described in B.2, with a single stem process. The models are conditioned on the function in Equation (37).

Table 13: MAE for ZINC250K conditioned on single properties.

|  | parp1 | fa7 | 5ht1b | braf | jak2 |
|---|---|---|---|---|---|
| GDSS | 5.56 | 4.76 | 5.78 | 5.73 | 5.98 |
| MOOD ood=0.04 | 5.42 | 4.33 | 5.52 | 5.37 | 5.10 |
| MOOD ood=0.01 | 5.41 | 4.33 | 5.52 | 5.36 | 5.09 |
| Twigs | **5.38** | **4.30** | **5.43** | **5.28** | **5.01** |

**Results.** In Table 13, we report the mean MAE values over multiple runs computed from the generated molecules using the pre-trained classifiers from [45]. We can observe that the `Twigs` consistently achieves a lower error, demonstrating an improved control over generating molecules with the desired target proteins.

**Runtime.** We have incorporated the runtime for molecule generation at inference time for a large-scale dataset (ZINC250K) as for Section 4.3, in Table 14. A comparison with MOOD [45] indicates that our model incurs a certain overhead, as anticipated. However, it demonstrates improved alignment when generating conditional molecules.

Table 14: Runtime for inference on molecule generation.

| model | Seconds per molecule |
|---|---|
| Twigs | 0.378 |
| MOOD | 0.267 |

## D   Experimental details

### D.1   Computational resources

All experiments are performed with GPUs, Nvidia A100 or v100.

### D.2   Models details

We follow the data splits from Huang et al. [28] for 4.1, 4.2, the ones from Lee et al. [45] for 4.3, and the data splits from Jo et al. [37] for 4.4. We use Adam optimizers on all experiments.

For Sections 4.1 and 4.2 we follow the same hyperparameters from Huang et al. [28]. For Section 4.3 we follow the hyperparameters from Lee et al. [45], for the MOOD baseline, we explore OOD coefficients between 0.01 and 0.09. For Section 4.4 we follow the hyperparameters from Jo et al. [37].

## E   Additional Related Works

This section extends the discussion presented in Section 2 by exploring additional related works in the field. In Table 15 we summarise related methods including score-sdes, hierarchical models (not necessarily conditional), and hierarchical conditional models.

**Conditional molecular diffusion.** Guidance techniques have also been adopted in conditional molecule generation settings: in the context of classifier-free approaches, Hoogeboom et al. [26] proposes an equivariant approach based on DDPM for 3D molecules; Huang et al. [28] explores attention mechanisms within SGM models; and Xu et al. [82] investigates DDPMs in latent space settings.

In terms of classifier-based guidance, Bao et al. [3] incorporate energy guidance into a diffusion model by leveraging a stochastic differential equation; Vignac et al. [75] provide a DDPM coupled with a classifier over quantum molecular properties; and Lee et al. [45] operate over a pre-trained SGM and train an additional predictor for fine-tuning the desired protein target properties.

Table 15: Comparison with related works.

| Method | Score-based SDE | Hierarchical modeling | Hierarchical conditional diffusion |
|---|---|---|---|
| EDM [26] | ✗ | ✗ | ✗ |
| EEGSDE [3] | ✓ | ✗ | ✗ |
| Digress [75] | ✗ | ✗ | ✗ |
| HierVAE [34] | ✗ | ✓ | ✗ |
| GraphGuide [72] | ✗ | ✓ | ✗ |
| GeoLDM [82] | ✗ | ✗ | ✗ |
| HierGraph [57] | ✗ | ✓ | ✗ |
| JODO [28] | ✓ | ✗ | ✗ |
| Twigs (this work) | ✓ | ✓ | ✓ |

**Guidance methods.** Recent works utilize multiple diffusion processes: cascaded diffusion [25], provides a flow for each resolution, and GDSS [37] has a joint system of diffusion processes one for nodes and the other for edge features, but it does not cover mechanisms for conditional generation. Tseng et al. [71] define a hierarchy of branching points within a single diffusion flow.

**Other Diffusion methods for Graphs.** Other works related to ours focus on hierarchical diffusion processes [7], diffusion applied to protein backbones [69], geometry-based models [59, 82], and autoregressive models [41]. In the realm of stochastic differential equation (SDE)-based approaches, the literature includes bridge methods [38], permutation invariance [27], torsional modeling [36], and docking [6]. Additionally, [63] introduces the ConfGF approach, estimating gradient fields of atomic coordinates, while [80] proposes a method steering the training of diffusion-based generative models using physical and statistical prior information.

**Autoencoder-Based graph models.** This category includes works employing autoencoders, such as retrieval-based models [77, 12], scaffold modeling [50], link design [29], and coarse-grain modeling [76]. Notably, [54] proposes a reaction-embedded and structure-conditioned variational autoencoder, while [42] defines the concept of principal subgraphs, relevant to informative patterns within molecules.

**Conditional Diffusion.** In the realm of diffusion generative models, several noteworthy approaches have been developed to enhance their performance and versatility. Du et al. [10] introduce an energy-based parameterization of diffusion models, allowing the integration of novel compositional operators and Metropolis-corrected samplers. Building on this, He et al. [22] contribute a training-free conditional generation framework, leveraging pretrained diffusion models focusing on the manifold hypothesis to refine guided diffusion steps and introduce a shortcut algorithm. Meanwhile, Meng et al. [51] employ a stochastic differential equation (SDE) in synthesizing realistic images, iterating through denoising steps guided by a pretrained diffusion model.

In a different vein, Song et al. [66] propose guiding denoising diffusion models with general differentiable loss functions in a plug-and-play manner, facilitating controllable generation without additional training. Addressing the challenge of inferring high-dimensional data within the context of diffusion models, Graikos et al. [17] present a model consisting of a prior and an auxiliary differentiable constraint. Dinh et al. [9] tackle diversity and adversarial effects in classifier guidance for diffusion generative models by allowing relevant classes' gradients to contribute to shared information construction during noisy early sampling steps. Furthermore, Song et al. [65] put forth a method for estimating conditional scores without additional training. Lastly, Ouyang et al. [56] propose the Contrastive-Guided Diffusion Process (Contrastive-DP), integrating contrastive loss to guide the diffusion model in data generation. These diverse contributions collectively advance the field by addressing various challenges and expanding the capabilities of diffusion generative models.

