# OpenReview forum: "Diffusion Twigs with Loop Guidance for Conditional Graph Generation"
_NeurIPS.cc/2024/Conference — NeurIPS 2024 poster_

### Official Review · Reviewer_Gpe5 · 2024-07-04

**Soundness:** 3
**Presentation:** 3
**Contribution:** 2
**Rating:** 5
**Confidence:** 3

**Summary:**

The paper presents Twigs, a score-based diffusion framework for enhancing conditional generation tasks. It includes a central trunk diffusion process for graph structure and stem processes for graph properties. The innovative loop guidance strategy manages information flow between trunk and stem processes. Experiments show Twigs significantly outperforms contemporary baselines, especially in tasks like inverse molecular design.

**Strengths:**

1. A new guidance strategy that explicitly models the coupling effects between condition and data.
2. Impressive experimental results.

**Weaknesses:**

1. The main drawback of the proposed method is the limited flexibility. The loop guidance requires the joint training of $s_{\theta}$ and $s_{\phi}$ and it cannot provide guidance for unseen conditions.
2. Some of the experimental settings and evaluation pipeline are unclear. Please at lease specify them in the appendix.
3. Minor: Please put back the guidelines of paper checklist.

**Questions:**

1. How will the sampling step size affect the generation? Can we use different step sizes when sampling $y_s$ and $y_i$?
2. Classifier guidance helps the diffusion model to generalize to new conditions. Can Twigs provide guidance of unseen conditions by, for example, separating the training of $s_{\theta}$ and $s_{\phi}$?
3. How will the incorporation of multiple $s_{\phi}$ influence the total training time of Twigs? Can you provide the training time comparison as well?

**Limitations:**

The limitations mentioned in the manuscript are more like future direction of applying the similar strategy to other domains. Please provide more discussion if possible.

---

> ### Author Rebuttal · Authors · 2024-08-06
>
> Thank you very much for your many excellent suggestions! We have acted on all of these, and additionally, address all your questions and comments below.
>
> **RE: Twigs cannot guide unseen conditions (W1) (Q2)**
>
> Thanks for raising this point. Based on your suggestion we have added an experiment to tackle the issue that you raised. The Twig framework is very flexible, and it can be trained with unconditional setting, and combined with a classifier, to achieve generalization, as you mentioned. We provide here an example.
>
> Specifically, we first train an unconditional version of Twigs, which consists of a diffusion model over $k$ graph properties. Secondly, we train a separate discriminator for the new property to achieve prediction guidance. This idea is the same as in classifier guidance, and in MOOD (Lee et al 2023), which allows the model to generalize to properties not seen during the training of the diffusion model.
>
> The table below shows the unconditional version of Twigs trained with three properties (excluding the unseen property) and a separate property predictor (for the unseen property), on the `community-small` dataset. The results show that Twigs receives benefits from having the diffusion over multiple properties.
>
> | Property      | MAE values |          |         |      |
> | ------------- | ---------- | -------- | ------- | ---- |
> |               | Twigs p=3  | MOOD     | Digress | GDSS |
> | Density       | **2.12**   | **2.12** | 2.34    | 2.95 |
> | Clustering    | **9.94**   | 11.3     | 10.6    | 12.1 |
> | Assortativity | **15.8**   | 16.7     | 17.8    | 19.6 |
> | Transitivity  | **8.68**   | 8.76     | 9.42    | 11.4 |
>
>
> **RE: experimental settings and evaluation pipeline are unclear (W2)**
>
> Thank you, based on your comment, we have added the hyperparameters used in the experiments, which we will integrate into the main text. In particular, we will clarify the following settings.
> For Sections 4.1 and 4.2 we follow the same hyperparameters from Huang et al (2023).
> For Section 4.3 we follow the hyperparameters from Lee et al (2023), for the MOOD baseline, we explore the OOD coefficient between $0.01$ and $0.09$.
> For Section 4.4 we follow the hyperparameters from Jo et al (2022).
>
>
> **Guidelines checklist (W3):**
> Yes, we will bring it back.
>
>
> **RE: how step sizes affect generation. Q1**
>
> For step-size Sections 4.1 and 4.2, we follow the same hyperparameters from Huang et al (2023), including step-sizes of the sampling procedure. For Section 4.3 we follow the hyperparameters from Lee et al (2023), which leverages GDSS.
>
>
> **RE: can we have different main and secondary process stepsizes. Q1**
> To generalize to unseen properties, we do not need to separate the diffusion, we show above that we can train a separate predictor on unseen properties. Therefore we can maintain the same stepsizes in the diffusion flows.
>
>
>
> **RE: provide the training time comparison in multiple diffusion (Q3)**
>
> We show the impact of multiple diffusion flows on the community-small and Enzymes datasets. Specifically,  we report the average time (in seconds) for the training of a single epoch for Twigs with one and three secondary diffusion flows.
> We observe that our models encounter a small overhead compared to GDSS and Digress, however, we believe it is a good tradeoff for performance.
>
>
> | Dataset         | Twigs p=1 | Twigs p=3 | GDSS   | Digress |
> | --------------- | --------- | --------- | ------ | ------- |
> | Community-small | 0.2747    | 0.2997    | 0.2294 | 0.2382  |
> | Enzymes         | 4.8669    | 5.0304    | 4.8260 | 4.8451  |
>
>
> **Final comment**
>
> We are grateful for your thoughtful review. We hope our response has addressed your questions and concerns, and will appreciate the same being reflected in your stronger support for this work.

---

> > ### Comment · Reviewer_Gpe5 · 2024-08-09
> > **Thank you for the rebuttal**
> >
> > Thank you for the response and the additional experimental results. The comparison of per epoch training time suggests a slight overhead. Since different models will have different numbers of optimal training epochs, can you also include the total training time in comparison?
> >
> > Regarding the experimental settings, please explicitly write out the dataset preprocessing, training/test data splitting, evaluation metrics, and the choice of hyperparameters in addition to the citations in later revision.

---

> > > ### Author Response · Authors · 2024-08-12
> > >
> > > Dear Reviewer Gpe5, thank you for requesting additional information, which helps to highlight the advantages of our method. As shown in the table below, the overall training times indicate that the overhead of our model is negligible. To ensure fairness, we trained all models for 5,000 epochs on both datasets.
> > >
> > > | Dataset         | Twigs p=1 | Twigs p=3 | GDSS   | Digress |
> > > | --------------- | --------- | --------- | ------ | ------- |
> > > | Community-small | 0h 22m    | 0h 24m    | 0h 19m | 0h 20m  |
> > > | Enzymes         | 6h 45m    | 6h 59m    | 6h 42m | 6h 43m  |
> > >
> > >
> > > **RE: experimental settings**. Thanks for making the request explicit which helps improving our work. We will update all the experimental details of the data and hyperparameters systematically in the main text.

---

> ### Comment · Reviewer_Gpe5 · 2024-08-13
> **Thank you for the reply**
>
> Thank you for the additional results. Given the above discussion, seems the proposed method provides limited performance improvement while introducing small computational overheads. Can you elaborate more on what is the advantage of the proposed method compared to MOOD and JODO?

---

> > ### Author Response · Authors · 2024-08-13
> >
> > Thanks for the opportunity to clarify the strengths of our method compared to JODO and MOOD.
> >
> > **Novelty of Twigs**
> >
> > The idea of our method (Twigs) stems from the goal of uncovering the intricate relationships between the graph structure and each of the target properties. As a result, we define a hierarchy of diffusion flows, aimed to improve the representational power of the neural network. The experiments show that the orchestration of our novel hierarchical model leads to the learning of richer conditional representations, leading to improvements when compared to previous guidance methods including MOOD and JODO.
> >
> > Specifically, we define two types of diffusion processes with the following roles: (1) multiple *Stem* processes, which aim to unravel the interactions between the graph structure and single properties with the networks $s_{\phi_i}$, and (2) the *Trunk* process, which orchestrates the combination of the graph structure score from $s_\theta$ with the stem process contributions from $s_{\phi_i}$. Twigs resembles a cycle going from the stem process into the trunk process, which we name *loop guidance*.
> >
> >
> >
> > **Summary of computational experiments**
> >
> > In practice, our method is able to outperform JODO for generation over single and multiple target property, and MOOD on both property optimization and multi-property conditional generation.
> >
> > **Comparison with JODO**
> >
> > First, we report the MAE results for single quantum properties from the `QM9` dataset (lower the better). Twigs outperforms JODO on all properties.
> >
> > | Model     | $C_v$               | $\mu$               | $\alpha$          | $\Delta \epsilon$ | $\epsilon_\text{HOMO}$ | $\epsilon_\text{LUMO}$ |
> > | --------- | ------------------- | ------------------- | ----------------- | ----------------- | ---------------------- | ---------------------- |
> > | JODO      | 0.581 (± 0.001)     | 0.628 (± 0.003)     | 1.42 (± 0.01)     | 335 (± 3)         | 226 (± 1)              | 256 (± 1)              |
> > | **Twigs** | **0.559 (± 0.002)** | **0.627 (± 0.001)** | **1.36 (± 0.01)** | **323 (± 2)**     | **225 (± 1)**          | **244 (± 3)**          |
> >
> > Secondly, we show that the advantage holds for multiple properties. We consider three molecular properties ($\alpha$, $\mu$, $\Delta \epsilon$) for the `QM9` dataset. Our model (Twigs) consistently achieves significantly lower MAE values compared to JODO, underscoring the enhanced accuracy and reliability of our predictions.
> >
> > | Model | MAE values           |                      |                   |
> > | ----- | -------------------- | -------------------- | ----------------- |
> > |       | $\alpha$             | $\mu$                | $\Delta \epsilon$ |
> > | JODO  | 2.749 $\pm$ 0.03     | 1.162 $\pm$ 0.04     | 717 $\pm$ 5       |
> > | Twigs | **2.544 $\pm$ 0.05** | **1.094 $\pm$ 0.02** | **640 $\pm$ 3**   |
> >
> >
> >
> >
> > **Comparison results with MOOD**
> >
> > First we report the Novel top 5% docking scores on `ZINC250k` (higher is better). This metric considers multiple constraints over QED, SA, and tanimoto similarity. Our model achieves improved scores in 4 out of 5 cases.
> >
> > | Model | parp1                | fa7                 | 5ht1b                | braf                 | jak2                |
> > | ----- | -------------------- | ------------------- | -------------------- | -------------------- | ------------------- |
> > | MOOD  | 10.409 (± 0.030)     | 7.947 (± 0.034)     | 10.487 (± 0.069)     | **10.421 (± 0.050)** | 9.575 (± 0.075)     |
> > | Twigs | **10.449 (± 0.009)** | **8.182 (± 0.012)** | **10.542 (± 0.025)** | 10.343 (± 0.024)     | **9.678 (± 0.032)** |
> >
> >
> > Secondly, we report the Novel hit ratio on `ZINC250k` (higher is better). The metric represent the fraction of hit molecules constrained on tanimoty similarity. The richer representation of our model lead to improvements on 4 out of 5 properties.
> >
> > | Model | parp1               | fa7                 | 5ht1b                | braf                | jak2                |
> > | ----- | ------------------- | ------------------- | -------------------- | ------------------- | ------------------- |
> > | MOOD  | 3.400 (± 0.117)     | 0.433 (± 0.063)     | 11.873 (± 0.521)     | **2.207 (± 0.165)** | 3.953 (± 0.383)     |
> > | Twigs | **3.733 (± 0.081)** | **0.900 (± 0.012)** | **16.366 (± 0.029)** | 1.933 (± 0.023)     | **5.100 (± 0.312)** |
> >
> >
> >
> > Finally, we report MAE values for graph generation conditioned on three properties on the `community-small` dataset (lower is better). Twigs achieves lower errors on all the considered target properties.
> >
> > | Model | MAE Density | MAE Clustering | MAE Assortativity |
> > | ----- | ----------- | -------------- | ----------------- |
> > | MOOD  | 2.53        | 11.4           | 17.3              |
> > | Twigs | **2.27**    | **10.6**       | **16.1**          |
> >
> >
> >
> > Please, let us know if we have addressed your concerns.

---

> > > ### Comment · Reviewer_Gpe5 · 2024-08-13
> > >
> > > I appreciate the authors for their rebuttal efforts and I have increased my score to 5. Please include the training time comparison (preferably with both MOOD and JODO) in the later revision.

---

### Official Review · Reviewer_8xx4 · 2024-07-10

**Soundness:** 3
**Presentation:** 3
**Contribution:** 3
**Rating:** 6
**Confidence:** 3

**Summary:**

This study proposed a conditional generative model with guided diffusion. The authors introduce a new mechanism called loop guidance to include conditions. Empirical analysis includes small molecule generation on a diverse of datasets and properties, with both quantitative evaluation and visualizations.

**Strengths:**

1. The paper presents a well-structured framework for conditional generation with guided diffusion, including a neat summary of relevant methods in this field.
2. The presentation of the methods and results is clear and easy to follow.
3. The authors provide extensive experimental comparisons, offering a thorough evaluation of the proposed method against various baselines.

**Weaknesses:**

The weaknesses mainly arise from the insufficient empirical evidence:

1. No comparison has been provided regarding the computational cost, making it unclear how efficient the proposed method is relative to others.
2. Although the authors claimed that conditional generation is a fast-developing research field, the baseline methods used for comparison are generally from before 2023, which may not reflect the current SOTA.
3. No ablation study or hyperparameter selection/analysis has been reported.
4. Some terminologies could be introduced and analyzed more carefully (see question 1 below).

**Questions:**

1. What are the meanings of the properties $C_v$, $\mu$, $\alpha$, $\Delta\epsilon$, $\epsilon_{\text{LUMO}}$, and $\epsilon_{\text{HOMO}}$? Why are they important impact factors for molecule generation? Are they practically easy to obtain? Since these properties are used as guides when generating small molecules, do the generated molecules actually have properties close to the guide?
2. What is the relationship and difference between the proposed method and other recent works on guided diffusion, such as https://arxiv.org/pdf/2406.01572, https://arxiv.org/abs/2305.20009, and https://openreview.net/pdf?id=8NfHmzo0Op? (Note: I'm not requiring a quantitative comparison, especially for the first one, which was uploaded after the conference submission deadline. However, for the latter two papers, as they have been published for more than 3 months, I believe they should be included in the paper.)
3. How should the top row of Figure 2 be interpreted? Are they different samples from the same guidance, or are they the same sample shown from different views?
4. Similarly, it is not clear how to interpret the results in Figure 3 (and this figure was not referenced in the main text). Are all the samples generated based on the same input properties? How can the legitimacy of these generations be validated? What does the ground-truth molecule look like? To what level they are better than the generation performance of the baseline methods?

**Limitations:**

The authors listed one limitation of the current work as not generalizing the framework to other types of data (text, image). No additional analysis on the broader impact has been provided.

---

> ### Author Rebuttal · Authors · 2024-08-06
>
> Many thanks!
>
> **RE: computational cost  (W1)**
>
> In Table 12 of paper, we present the inference times of our model compared to MOOD, showing that while we encounter a small overhead, we achieve superior performance.
>
> In addition, we report below the average time (in seconds) required to train a single epoch for the `community-small` and `Enzymes`. For Twigs, we consider one property (density).
>
>
> | Dataset         | Twigs p=1 | GDSS   | Digress |
> | --------------- | --------- | ------ | ------- |
> | Community-small | 0.2747    | 0.2294 | 0.2382  |
> | Enzymes         | 4.8669    | 4.8260 | 4.8451  |
>
>
>
> **baseline methods (W2)**
>
> We provide a list of the recent methods (after 2023) in our experiments.
> In Sections 4.1 and 4.2, we compare Twigs with the following methods on QM9: TEDMol (Luo et al 2024), Jodo (Huang et al 2023), EEGSDE (Bao et al 2023), GeoLDM (Xu et al 2023), EquiFM (Song et al 2023). In Section 4.3, we compare with Lee et al (2023). In Section 4.4, we compare with Lee et al (2023), and Vignac et al (2023) on the `community-small` dataset.
>
>
>
>
>
> **Hyperparameters (W3)**
>
> Here we report the hyperparameters used in the experiments.
> For Sections 4.1 and 4.2 we follow the same hyperparameters from Huang et al (2023).
> For Section 4.3 we follow the hyperparameters from Lee et al (2023), for the MOOD baseline, we explore the OOD coefficient between $0.01$ and $0.09$.
> For Section 4.4 we follow the hyperparameters from Jo et al (2022).
>
>
> **Ablation study (W3)**
>
> We have added a new study to understand the impact of multiple properties over the `community-small` dataset. We report the results in the "Additional experiments with multiple properties" part of the [global comment](https://openreview.net/forum?id=fvOCJAAYLx&noteId=SQNeH19zc9).
>
>
> **Refs**
>
> Huang et al (2023). Learning joint 2d & 3d diffusion models for complete molecule generation.
>
> Lee et al (2023). Exploring chemical space with score-based out-of-distribution generation.
>
> Jo et al (2022). Score-based generative modeling of graphs via the system of stochastic differential equations.
>
> Bao et al 2023. Equivariant Energy-Guided SDE for Inverse Molecular Design.
>
> Luo et al 2024. Text-guided diffusion model for 3d molecule generation.
>
> Xu et al 2023. Geometric latent diffusion models for 3d molecule generation.
>
> Song et al 2023. Equivariant flow matching with hybrid probability transport for 3d molecule generation.
>
> Vignac et al 2023. Digress: Discrete denoising diffusion for graph generation
>
>
>
> **properties close to the guide (Q1)**
>
> The generated molecules' properties closely align with the target values, as evidenced by the MAE values presented in Tables 3, 5, and 8. The MAE quantifies the deviation between the desired ground truth properties and those of the generated molecules.
>
>
> **meanings of the properties (Q1)**
>
> The mentioned properties are standard quantum properties used in the QM9 dataset for modeling molecules (Hoogeboom et al 2022). The molecule properties are obtained from the data.
>
> - $\alpha$ Polarizability: Tendency of a molecule to acquire an electric dipole moment when subjected to an external electric field.
>
> - $\epsilon_{\text{HOMO}}$: Highest occupied molecular orbital energy.
>
> - $\epsilon_{\text{LUMO}}$: Lowest unoccupied molecular orbital energy.
>
> - $\Delta \epsilon$ Gap: The energy difference between HOMO and LUMO.
>
> - $\mu$: Dipole moment.
>
> - $C_v$: Heat capacity at 298.15K.
>
>
> **Refs:**
>
> Hoogeboom et al 2022. Equivariant Diffusion for Molecule Generation in 3D. ICML.
>
>
> **related works (Q2)**
>
> Our approach differs from the mentioned methods in two key ways, which we will elaborate on in the related works section of our paper.
>
> **Framework**: The first two studies (Nisonoff et al. 2024, Gruver et al. 2023) are based on discrete frameworks. The third study (Klarner et al. 2024) adopts a plug-and-play approach using a diffusion model, which is agnostic to the underlying framework. Differently from the above, our method operates within a continuous framework utilizing stochastic differential equations (SDEs).
>
> **Contributions**:  Nisonoff et al. 2024 primary contribution is to provide a framework for diffusion guidance within discrete spaces. On the other hand, the main contribution of Gruver et al. (2023), addresses the challenges of optimizing discrete sequences by a novel sampling technique.
> Klarner et al. 2024 propose to learn two models one diffusion without labels and an additional discriminator model with property labels.
> Our contribution is separate from all the above methods, as we uniquely leverage multiple diffusion flows, defined over two distinct types of flows within a hierarchical structure. Specifically, we define a primary flow for capturing structure and a secondary flow for modeling properties.
>
>
> **Refs**
>
> Nisonoff et al (2024). Unlocking Guidance for Discrete State-Space Diffusion and Flow Models.
>
> Gruver et al (2023). Protein Design with Guided Discrete Diffusion.
>
> Klarner et al (2024). Context-Guided Diffusion for Out-of-Distribution Molecular and Protein Design.
>
>
>
> **Clarification on Figs (Q3-Q4)**
>
> Figure 2: Depits uncurated samples obtained via our model conditioned on $C_v$ property. The performance of conditioning is given in Table 3 First column ($C_v$). In addition, to validate the legitimacy, those molecules are representative of the "Molecule quality" results from Table 4.
>
> Figure 3. It shows uncurated samples of molecules produced under two desired properties ($C_v$ and $\mu$). The performance corresponding to Fig 3 is represented in Table 5 first two columns  ($C_v$ and $\mu$).
>
> Note that in this setup, our goal is not to reconstruct a specific molecule there could be multiple different molecules that have the desired properties and do not reconstruct the training set. In this sense, the target properties should be considered separately from the molecule.
>
> We would be grateful if this could be reflected in an increased score for our work. Thank you!

---

> > ### Author Response · Authors · 2024-08-11
> > **Follow up**
> >
> > Dear Reviewer 8xx4,
> >
> > Thanks again for your thorough review and constructive comments.
> >
> > Following up, we would greatly appreciate any updates or feedback you may have regarding our rebuttal.
> >
> > We would also appreciate any further comments and questions. Your constructive feedback can help us enhance the quality of our manuscript.
> >
> > Thank you and we look forward to your response.

---

> > > ### Comment · Reviewer_8xx4 · 2024-08-12
> > > **Reply to the Author Response**
> > >
> > > Thanks for your detailed response and sorry for the late reply. My concerns have been addressed and I would like to update my score to 6.

---

### Official Review · Reviewer_WGvA · 2024-07-11

**Soundness:** 2
**Presentation:** 3
**Contribution:** 2
**Rating:** 5
**Confidence:** 3

**Summary:**

This paper introduces Twigs, a new score-based diffusion model for graph generation conditioned on graph-level properties. It employs two diffusion processes, one for graph data (trunk process) and one for graph-level properties (stem process). The underlying generation process corresponds to a factorization of the joint distribution into the unconditional distribution of the graph data and conditionally independent distributions of graph properties given the graph data. Empirical studies across diverse benchmarks demonstrate the effectiveness of the proposed approach.

**Strengths:**

**S1:** The overall presentation is clear and easy to follow.

**S2:** Extensive empirical studies demonstrate the effectiveness of the method for conditional molecule and graph structure generation.

**Weaknesses:**

**W1**: The idea of hierarchical (conditional) diffusion model for graph generation has been explored in previous works. For example, GDSS [1] and EDGE [2] explore the factorization of the joint distribution of graph structure and node attributes for unconditional molecule generation. GraphMaker [3] explores this idea for node-label-conditioned generation of large attributed graphs.

**W2**: The assumption of the conditional independence of the multiple graph properties may be too strong.

**W3**: The empirical studies performed consider at most modeling two graph properties at a time and there is a lack of understanding in how the model performance changes as more graph properties are modeled simultaneously. In addition, there is a lack of understanding in how the model performance varies as the properties get more correlated.

**W4**: Some presentation details can be further clarified. For example:
- The terms "hierarchical modeling" and "hierarchical conditional diffusion" are used without a formal definition in table 1.
- I guess $y_{s,t}$ between L110 and 111 refers to $y_{s}$ at time $t$, but it is not formally defined.
- L106 says that $y_s$ is the primary variable (graph structure) and takes shape $\mathbb{R}^{N\times D}$. Does this consist of both adjacency matrix and node features like atom types?

[1] Jo et al. Score-based Generative Modeling of Graphs via the System of Stochastic Differential Equations.

[2] Chen et al. Efficient and Degree-Guided Graph Generation via Discrete Diffusion Modeling.

[3] Li et al. GraphMaker: Can Diffusion Models Generate Large Attributed Graphs?

**Questions:**

In addition to the issues mentioned in the "Weaknesses" section,

**Q1**: Why did you not include the results of EDM and TEDMol in table 4, as reported in [1]?

[1] Luo et al. Text-guided Diffusion Model for 3D Molecule Generation.

**Limitations:**

The paper assumes conditional independence between multiple graph properties, which does not necessarily hold in practice.

---

> ### Author Rebuttal · Authors · 2024-08-06
>
> Many thanks.
>
> **Re: related works (W1)**
>
> The suggested works, including GDSS (Jo et al. 2022), GraphMaker (Li et al. 2024), and EDGE (Chen et al. 2023), are relevant to our research and will be included in our paper. Below we report the differences with our method. Refer to the table of the global comment for a summary.
>
> First, these methods use different hierarchical notions in their frameworks. In EDGE and GDSS, edges and nodes are modeled symmetrically. In contrast, we introduce a unique hierarchical approach by learning multiple asymmetric diffusion flows. This distinction is crucial: while the mentioned methods use diffusion flows in the same roles, our method uses a secondary process to learn interactions with properties, feeding back into the main process to learn the graph structure.
>
> Secondly, our methodology is designed specifically for generating graphs with desired conditional properties, which the mentioned papers do not focus on. GDSS and EDGE do not consider conditional modeling, and while GraphMaker explores node-label conditioning, this differs from our goal of generating graphs constrained by specific properties. Our method learns a joint distribution of graph structures (nodes, edges) and properties, unlike the mentioned methods that only learn a joint distribution of nodes and edges.
>
>
> **RE: conditional independence (W2)**
>
> Assuming conditional independence among the properties $\alpha$, $\epsilon_{\text{HOMO}}$, $\epsilon_{\text{LUMO}}$, $\Delta \epsilon$, $\mu$, and $C_v$ given the molecular graph can simplify the modeling process. This assumption leverages the fact that the molecular graph captures the essential structural dependencies, allowing us to treat the properties as independent for computational efficiency and ease of interpretation, even if slight interdependencies exist.
>
>
> **RE: Modeling multiple properties (W3)**
>
> We show here two additional results on 2 and 3 properties. We study the models over the `community-small` dataset. Specifically, our method (Twigs) is trained using multiple secondary processes (one for each property).
>
> **Two properties**
>
> We show the MAE results for property pairs for density. Our model achieves the lowest MAE in both cases. We notice that while the properties may present some form of correlation, our model can achieve a good performance in generating the graphs with desired properties.
>
>
> | Model   | MAE Density | MAE Clustering |
> | ------- | ----------- | -------------- |
> | GDSS    | 2.95        | 13.3           |
> | Digress | 2.82        | 12.1           |
> | MOOD    | 2.43        | 12.0           |
> | Twigs   | **2.34**    | **11.0**       |
>
> | Model   | MAE Density | MAE Assortativity |
> | ------- | ----------- | ----------------- |
> | GDSS    | 2.61        | 19.8              |
> | Digress | 2.52        | 18.1              |
> | MOOD    | 2.40        | 17.2              |
> | Twigs   | **2.39**    | **16.7**          |
>
>
> **Three properties**
>
> The Table below shows that our method achieves the lowest MAE values (the lower the better) across all three required properties.
>
> | Model   | MAE Density | MAE Clustering | MAE Assortativity |
> | ------- | ----------- | -------------- | ----------------- |
> | GDSS    | 2.97        | 12.5           | 19.4              |
> | Digress | 2.65        | 11.2           | 18.2              |
> | MOOD    | 2.53        | 11.4           | 17.3              |
> | Twigs   | **2.27**    | **10.6**       | **16.1**          |
>
>
>
>
> **RE: Hierarchical conditional diffusion (W4.1)**
>
> Here's a clarification of "Hierarchical Conditional Diffusion" and the distinction from "unconditional hierarchical models", which will be integrated into the paper.
>
> In lines 39-43, we define "Hierarchical Conditional Diffusion" as follows: ".. rather than treating heterogeneous structural and label information uniformly within the hierarchy, we advocate for the co-evolution of multiple processes with distinct roles. These roles encompass a primary process governing structural evolution alongside multiple secondary processes responsible for driving conditional content."
>
> To distinguish our method from "unconditional" hierarchical models by Jin et al. (2020) and Qiang et al. (2023), we label those models as "Hierarchical Modeling" in Table 1. While they model hierarchical structures, our approach is unique in leveraging a hierarchy of branching diffusion processes for conditional generation based on desired properties.
>
> **References**
>
> Jin et al. 2020. Hierarchical generation of molecular graphs using structural motifs.
>
> Quiang et al. 2023. Coarse-to-fine: a hierarchical diffusion model for molecule generation in 3d.
>
>
> **RE: Variable $y_s$ (W4.2)**
> In the variable $y_{s,t}$ t indicates time. We will add to the main text.
>
>
> **RE: Dimension of graph (W4.3)**
>
> We have two cases:
>
> The 3D case in (Appendix B.1), we denote the variable $y_s$ as a 3D graph $G = (A, x, h)$, with node coordinates $x= (x^1, \ldots, x^N) \in \mathbb{R}^{N \times 3}$, node features $h = (h^1, \ldots,h^N) \in \mathbb{R}^{N \times d1}$, and edge information $A \in \mathbb{R}^{N \times N \times d2}$.
>
> The 2D case in Appendix B.2:  we denote $y_s$ as a 2D graph with $N$ nodes we consider the variable $y_s=(X,A)\in \mathbb{R}^{N\times F}\times\mathbb{R}^{N\times N}$, where $F$ is the dimension of the node features, $X\in\mathbb{R}^{N\times F}$ are node features, $A\in\mathbb{R}^{N\times N}$ is weighted adjacency matrix. We define the perturbed property $y_i \in \mathbb{R}$ and the (fixed) property $y_C \in \mathbb{R}$.
>
>
> **RE: EDM and Tedmol results (Q1)**
>
> Apologies for the misleading results presentation. Due to space constraints, we placed the results for EDM and Tedmol in Table 10 of Appendix C.1. We will move it to the main text. These methods were included in the appendix because they are less competitive compared to the other baselines.
>
>
> **Final comment**
>
> We hope our response addresses your concerns and reinforces your support for this work.

---

> ### Comment · Reviewer_WGvA · 2024-08-07
>
> Thank you for the detailed response, and I've read it carefully. Still, my main concern is the lack of studies on modeling >2 properties for real-world datasets like QM9, where the assumption of conditional independence can be insufficient.

---

> > ### Author Response · Authors · 2024-08-11
> >
> > Thank you for proposing the experiment, as it has further highlighted the strengths of our approach. The table below presents the results on the QM9 dataset, comparing three molecular properties ($\alpha$, $\mu$, $\Delta \epsilon$) for our model, Twigs, and the JODO method. Our model consistently achieves significantly lower MAE values across all three properties, underscoring the enhanced accuracy and reliability of our predictions.
> >
> > | Model | MAE values           |                      |                   |
> > | ----- | -------------------- | -------------------- | ----------------- |
> > |       | $\alpha$             | $\mu$                | $\Delta \epsilon$ |
> > | JODO  | 2.749 $\pm$ 0.03     | 1.162 $\pm$ 0.04     | 717 $\pm$ 5       |
> > | Twigs | **2.544 $\pm$ 0.05** | **1.094 $\pm$ 0.02** | **640 $\pm$ 3**   |

---

> > > ### Comment · Reviewer_WGvA · 2024-08-11
> > >
> > > Thank you for the update. I've increased my rating from 4 to 5. Still, you may want to briefly discuss this point in the Limitation section.

---

### Official Review · Reviewer_GMeF · 2024-07-12

**Soundness:** 3
**Presentation:** 3
**Contribution:** 3
**Rating:** 5
**Confidence:** 3

**Summary:**

This paper proposes a novel score-based diffusion framework called Twigs that incorporates multiple co-evolving flows to capture complex interactions and dependencies for enriching conditional generation tasks. It consists of a central trunk process and additional stem processes, coordinated by a loop guidance strategy during sampling. Extensive experiments on conditional graph generation demonstrate Twigs' strong performance gains over baselines, highlighting its potential for challenging generative tasks like inverse molecular design.

**Strengths:**

1. Based on my knowledge, the idea of incorporating multiple co-evolving diffusion processes proposed in this paper is very novel, especially in the field of graph generation.
2. The paper is well written and provides sufficient background information, such as Table 2, to help understand the differences between the method proposed in this paper and other classifier-based and classifier-free methods.
3. The experiments are very comprehensive and thorough, and the experimental results fully demonstrate the outstanding performance of the proposed method.

**Weaknesses:**

1. The content in Section 3 is written too generally and does not address the specific characteristics of graph data very well. Additionally, the mathematical symbols used appear somewhat disorganized.
2. I am a little bit confused about the the dimension graph structure $y_s \in \mathbb{R}^{N\times D}$ on top of Eq. (1). Does $D$ covers a lot more information, e.g. node coordinates (dim 3), node features (dim $d_1$), and  edge information (dim $N \times d_2$), as indicated in Appendix B.1?

**Questions:**

1. Is the external context $y_C$ one of the $k$ dependent graph properties $y_k$ in Eq. 2?

**Limitations:**

Not applied.

---

> ### Author Rebuttal · Authors · 2024-08-06
>
> Thank you so much for your thoughtful comments. We address all your concerns, as described below.
>
> **RE: Section 3 is written too generally and does not address the specific characteristics of the graph (W1)**
>
> We maintain the section in a general format to accommodate multiple cases, specifically one for 2D graphs and another for 3D graphs. We achieve the desired flexibility by introducing a variable $y_s$ that encompasses both node features and the adjacency matrix (2D case), as well as the coordinates (3D case). Below are the detailed descriptions of the $y_s$ variable.
>
> **RE: Dimension of the graph structure in Eqn (1) (W2)**
>
> We address two cases:
>
> **3D Case (Appendix B.1)**: We denote the variable $y_s$ as a 3D graph $G = (A, x, h)$, where node coordinates are represented as $x = (x^1, \ldots, x^N) \in \mathbb{R}^{N \times 3}$, node features as $h = (h^1, \ldots, h^N) \in \mathbb{R}^{N \times d_1}$, and edge information as $A \in \mathbb{R}^{N \times N \times d_2}$.
>
> **2D Case (Appendix B.2)**: Here, we denote $y_s$ as a 2D graph with $N$ nodes. The variable is defined as $y_s = (X, A) \in \mathbb{R}^{N \times F} \times \mathbb{R}^{N \times N}$, where $F$ represents the dimension of the node features. In this case, $X \in \mathbb{R}^{N \times F}$ are the node features, and $A \in \mathbb{R}^{N \times N}$ is the weighted adjacency matrix. Additionally, we define the perturbed property $y_i \in \mathbb{R}$ and the fixed property $y_C \in \mathbb{R}$.
>
>
> **RE: External Context $y_C$ and Dependent Graph Properties (Q1)**
>
> Indeed, the external context $y_C$ can be **one or more** of the $k$ dependent graph properties. Specifically, $y_C$ represents the graph property (or properties) upon which we are conditioning. For instance, in the context of conditional generative modeling for drug design, we might seek a molecule with a specific $\epsilon_{\text{LUMO}}$ value. In this scenario, $\epsilon_{\text{LUMO}}$ serves as the $y_C$ variable. The properties listed in Equation (2) encompass all relevant characteristics of the molecule, such as $\alpha$, $C_v$, $\epsilon_{\text{HOMO}}$, and others. Our model aims to perform multiple $k$ diffusion processes for each property, with each process conditioned on the  context $y_C$ and the property $y_k$.
>
>
> **Final Comment**
>
> Thank you so much for your constructive feedback. If you believe we sufficiently addressed your concerns, we would appreciate an increase in your score for this paper.

---

> > ### Author Response · Authors · 2024-08-11
> > **Follow up**
> >
> > Dear Reviewer GMeF,
> >
> > Thanks again for your thorough review and constructive comments.
> >
> > Following up, we would greatly appreciate any updates or feedback you may have regarding our rebuttal.
> >
> > We would also appreciate any further comments and questions. Your constructive feedback can help us enhance the quality of our manuscript.
> >
> > Thank you and we look forward to your response.

---

> > ### Comment · Area_Chair_qyUZ · 2024-08-13
> > **To reviewer GMeF**
> >
> > Dear Reviewer GMeF,
> >
> > Please let the authors know if your concerns have been addressed.
> >
> > PS (to authors): Graphs with adjacency matrices that contain edge features are tensors not 3D graphs.
> >
> > Thanks,
> > AC

---

> ### Comment · Reviewer_GMeF · 2024-08-13
>
> Thanks for your response. My questions are well-resolved. I will keep my score.

---

### Author Response · Authors · 2024-08-06
**Additional experiments**

**Additional experiments with multiple properties**

Some reviewers asked about performance with multiple properties. We report the results for two and three properties on the `community-small` dataset. Note that Twigs is on classifier-free guidance in those results.

First, we show the results for property pairs for density: (density, clustering) and (density, assortativity). Our model (Twigs) achieves the lowest MAE in both cases. While the properties may present some correlation, our model can generate the graphs with desired properties more accurately than the baselines.

| Model   | MAE Density | MAE Clustering |
| ------- | ----------- | -------------- |
| GDSS    | 2.95        | 13.3           |
| Digress | 2.82        | 12.1           |
| MOOD    | 2.43        | 12.0           |
| Twigs   | **2.34**    | **11.0**       |

| Model   | MAE Density | MAE Assortativity |
| ------- | ----------- | ----------------- |
| GDSS    | 2.61        | 19.8              |
| Digress | 2.52        | 18.1              |
| MOOD    | 2.40        | 17.2              |
| Twigs   | **2.39**    | **16.7**          |

Secondly, we report the results for the `community-small` dataset on three properties for generic graphs (Density, Clustering, and Assortativity). The Table below shows that our method achieves the lowest MAE values (the lower the better) across all three required properties. In other words, the Twigs model approximates the desired properties more closely than the baselines.

| Model   | MAE Density | MAE Clustering | MAE Assortativity |
| ------- | ----------- | -------------- | ----------------- |
| GDSS    | 2.97        | 12.5           | 19.4              |
| Digress | 2.65        | 11.2           | 18.2              |
| MOOD    | 2.53        | 11.4           | 17.3              |
| Twigs   | **2.27**    | **10.6**       | **16.1**          |

**Classifier-based guidance experiment**
Next, we report results with classifier-based guidance, as requested by a reviewer.  The table below shows the unconditional version of Twigs trained with three properties (excluding the unseen property - Density) and a separate property predictor (for the unseen property), on the `community-small` dataset. The results show that Twigs receives benefits from performing diffusion over multiple properties.

| Property      | MAE values |          |         |      |
| ------------- | ---------- | -------- | ------- | ---- |
|               | Twigs p=3  | MOOD     | Digress | GDSS |
| Density       | **2.12**   | **2.12** | 2.34    | 2.95 |
| Clustering    | **9.94**   | 11.3     | 10.6    | 12.1 |
| Assortativity | **15.8**   | 16.7     | 17.8    | 19.6 |
| Transitivity  | **8.68**   | 8.76     | 9.42    | 11.4 |



**Wall-clock training time results**

We report the average time for training a single epoch in seconds. GDSS achieves the lowest time, however, Twigs achieves better accuracy on generation.

| Dataset         | Twigs p=1 | Twigs p=3 | GDSS   | Digress |
| --------------- | --------- | --------- | ------ | ------- |
| Community-small | 0.2747    | 0.2997    | 0.2294 | 0.2382  |
| Enzymes         | 4.8669    | 5.0304    | 4.8260 | 4.8451  |

---

### Author Response · Authors · 2024-08-07
**Related works**

We thank the reviewers for their feedback, and the (senior) area chairs and program chairs for their service. We summarise below how we have addressed the key issues raised by the reviewers.

**Comparison with related works**

Some reviewers asked about the differences with some additional related works. We summarise the differences with our model Twigs in the following table.

While some of the works model diffusion with multiple flows (Jo et al 2022, Chen et al 2023, Li et al 2024), they symmetrically model nodes and edges. Our work instead considers the graph properties as well, as a secondary process, that branches from the main one of the graph structure.

In addition, while other guidance methods are related (Nisonoff et al 2024), (Gruver et al 2023) (Klarner et al 2024), they do not leverage multiple diffusion flows. Our method is the only one that leverages multiple diffusion flows hierarchically for conditional generation.



| Method                      | Conditional | Asymmetric | Multiple flows | Continuous (SDEs) |
| --------------------------- | ----------- | ---------- | -------------- | ----------------- |
| Jo et al. 2022 (GDSS)       | ✗           | ✗          | ✓              | ✓                 |
| Chen et al. 2023 (EDGE)     | ✗           | ✗          | ✓              | ✗                 |
| Li et al. 2024 (GraphMaker) | ✓           | ✗          | ✓              | ✗                 |
| Nisonoff et al (2024)       | ✓           | ✗          | ✗              | ✗                 |
| Gruver et al (2023)         | ✓           | ✗          | ✗              | ✗                 |
| Klarner et al (2024)        | ✓           | ✗          | ✗              | ✓                 |
| Twigs (ours)                | ✓           | ✓          | ✓              | ✓                 |




**References:**

Jo et al (2022). Score-based generative modeling of graphs via the system of stochastic differential equations.

Chen et al. 2023. Efficient and Degree-Guided Graph Generation via Discrete Diffusion Modeling. ICML.

Li et al. 2024. GraphMaker: Can Diffusion Models Generate Large Attributed Graphs? Arxiv preprint.

Nisonoff et al (2024). Unlocking Guidance for Discrete State-Space Diffusion and Flow Models. Arxiv preprint.

Gruver et al (2023). Protein Design with Guided Discrete Diffusion. Neurips.

Klarner et al (2024). Context-Guided Diffusion for Out-of-Distribution Molecular and Protein Design.

---

### Decision · Program_Chairs · 2024-09-25

**Decision:**

Accept (poster)

**Comment:**

The paper presents Twigs, which aims to enhance the representation power of graph generators by focusing on the relationships between graph structure and target properties. Twigs introduces a hierarchy of diffusion flows, comprising multiple Stem processes and a Trunk process. The Stem processes focus on disentangling the interactions between the graph structure and individual properties, while the Trunk process orchestrates these contributions to generate conditional representations. The experimental results demonstrate that Twigs outperforms existing guidance methods, such as MOOD and JODO, in diffusion-based graph generation.

**Strengths:**
1. Reviewers liked the hierarchical diffusion processes, which is innovative and addresses the challenge of capturing complex interactions between graph structure and target properties.
2. The loop guidance mechanism, which iteratively refines the contributions from Stem to Trunk processes, is well-motivated.
3. The method's performance improvements over existing approaches, particularly MOOD and JODO, are OK.

**Weaknesses:**
1. The writing in the main paper needs some improvement. Reviewers initially provided negative feedback due to the lack of detail in Section 3 and incomplete information about the method in Section 4. The rebuttal successfully addressed these concerns, and these clarifications should be incorporated into the main text. My recommendation assumes the authors will make the necessary changes to the paper to improve its readability.
2. The scalability on the number of properties is a limitation of the method that needs to be better discussed in the main paper.
3. The comparison with baseline methods was less-than-ideal in the original version. Please add the new experiments to the paper.

**Recommendation:**
Overall, Twigs represents a good contribution to score-based diffusion graph generation, with multiple co-evolving flows to capture complex interactions and dependencies for enriching conditional generation tasks. With some additional clarity on the theoretical aspects and computational considerations, this work could have greater impact. I recommend acceptance under the assumption the authors will do the promised improvements to the paper (i.e., camera-ready will take time and effort). If the paper is accepted, I will check to see if the authors made the promised changes.